

# Nitrogen oxides in the free troposphere: Implications for tropospheric oxidants and the interpretation of satellite NO₂ measurements

Viral Shah[1,a], Daniel J. Jacob[1,2], Ruijun Dang[1], Lok N. Lamsal[3,4], Sarah A. Strode[3,5], Stephen D. Steenrod[3,4], K. Folkert Boersma[6,7], Sebastian D. Eastham[8,9], Thibaud M. Fritz[8], Chelsea Thompson[10,11], Jeff Peischl[10,11], Ilann Bourgeois[10,11,b], Ilana B. Pollack[12], Benjamin A. Nault[13], Ronald C. Cohen[14,15], Pedro Campuzano-Jost[16,17], Jose L. Jimenez[16,17], Simone T. Andersen[18], Lucy J. Carpenter[18], Tomás Sherwen[18,19], Mat J. Evans[18,19]

[1] Harvard John A. Paulson School of Engineering and Applied Sciences, Harvard University, Cambridge, MA 01238, USA
[2] Department of Earth and Planetary Sciences, Harvard University, Cambridge, MA 02138, USA
[3] Atmospheric Chemistry and Dynamics Laboratory, NASA Goddard Space Flight Center, Greenbelt, MD 20771, USA
[4] University of Maryland Baltimore County, Baltimore, MD 21250, USA
[5] GESTAR II, Morgan State University, Baltimore, MD 21251, USA
[6] Royal Netherlands Meteorological Institute (KNMI), De Bilt, the Netherlands
[7] Wageningen University, Wageningen, the Netherlands
[8] Laboratory for Aviation and the Environment, Department of Aeronautics and Astronautics, Massachusetts Institute of Technology, Cambridge, MA 02139, USA
[9] Joint Program on the Science and Policy of Global Change, Massachusetts Institute of Technology, Cambridge, MA 02139, USA
[10] NOAA Chemical Sciences Laboratory, Boulder, CO 80305, USA
[11] Cooperative Institute for Research in Environmental Sciences, University of Colorado Boulder, Boulder, CO 80309, USA
[12] Department of Atmospheric Sciences, Colorado State University, Fort Collins, CO 80523, USA
[13] Center for Aerosols and Cloud Chemistry, Aerodyne Research, Inc., Billerica, MA 01821, USA
[14] Department of Earth and Planetary Science, University of California Berkeley, Berkeley, CA 94720, USA
[15] Department of Chemistry, University of California Berkeley, Berkeley, CA 94720, USA
[16] Cooperative Institute for Research in Environmental Sciences, University of Colorado, Boulder, CO 80309, USA
[17] Department of Chemistry, University of Colorado, Boulder, CO 80309, USA
[18] Wolfson Atmospheric Chemistry Laboratories, Department of Chemistry, University of York, York, YO10 5DD, UK
[19] National Centre for Atmospheric Science, University of York, York YO10 5DD, UK.
[a] Now at Global Modeling and Assimilation Office, NASA Goddard Space Flight Center, Greenbelt, MD 20771, USA, and Science Systems and Applications, Inc., Lanham, MD 20706, USA
[b] Now at Extreme Environments Research Laboratory, École Polytechnique Fédérale de Lausanne Valais Wallis, Sion, Switzerland, and Plant Ecology Research Laboratory, École Polytechnique Fédérale de Lausanne, Lausanne, Switzerland.

*Correspondence to*: Viral Shah (vshah@seas.harvard.edu)

**Abstract.** Satellite-based retrievals of tropospheric NO₂ columns are used to infer NOₓ (≡NO+NO₂) emissions at the surface. These retrievals rely on model information for the vertical distribution of NO₂. The free tropospheric background above 2 km is particularly important because the sensitivity of the retrievals increases with altitude. Free tropospheric NOₓ also has a strong effect on tropospheric OH and ozone concentrations. Here we use observations from three aircraft campaigns (SEAC⁴RS, DC3, and ATom) and four atmospheric chemistry models (GEOS-Chem, GMI, TM5, and CAMS) to evaluate the model capabilities for simulating background NOₓ and attribute this background to sources. NO₂ measurements over the southeast



US during SEAC$^4$RS and DC3 show increasing concentrations in the upper troposphere above 10 km, which is not replicated by GEOS-Chem although the model is consistent with the NO measurements. Using concurrent NO, NO$_2$ and ozone observations from a DC3 flight in a thunderstorm outflow, we show that NO$_2$ measurements in the upper troposphere are

biased high, plausibly due to interference from thermally labile NO$_2$ reservoirs, such as peroxynitric acid (HNO$_4$) and methyl peroxy nitrate (MPN). We find that NO$_2$ concentrations calculated from the NO measurements and NO-NO$_2$ photochemical steady state (PSS) are more reliable to evaluate the vertical profiles of NO$_2$ in models. GEOS-Chem reproduces the shape of the PSS-inferred NO$_2$ profiles throughout the troposphere for SEAC$^4$RS and DC3 but overestimates NO$_2$ concentrations by about a factor of 2. The model underestimates MPN and alkyl nitrate concentrations, suggesting missing organic NO$_x$

chemistry. On the other hand, the standard GEOS-Chem model underestimates NO observations from the ATom campaigns over the Pacific and Atlantic Oceans, indicating a missing NO$_x$ source over the oceans. We find that we can account for this missing source by including in the model the photolysis of particulate nitrate on sea salt aerosols at rates inferred from laboratory studies and field observations of nitrous acid (HONO) over the Atlantic. The average NO$_2$ column density for the ATom campaign in the GEOS-Chem simulation is $2.4 \times 10^{14}$ molec cm$^{-2}$ with particulate nitrate photolysis and $1.5 \times 10^{14}$ molec

cm$^{-2}$ without, compared to $1.9 \times 10^{14}$ molec cm$^{-2}$ in the observations (using PSS NO$_2$) and $1.4$–$2.4 \times 10^{14}$ molec cm$^{-2}$ in the GMI, TM5 and CAMS models. We find from GEOS-Chem that lightning is the main primary NO$_x$ source in the free troposphere over the tropics and southern midlatitudes, but aircraft emissions dominate at northern midlatitudes in winter and in summer over the oceans. Particulate nitrate photolysis increases ozone concentrations by up to 5 ppbv in the free troposphere in the northern extratropics in the model, which would largely correct the low model bias relative to ozonesonde observations. Global

tropospheric OH concentrations increase by 19%. The contribution of the free tropospheric background to the tropospheric NO$_2$ columns observed by satellites over the contiguous US increases from 25% in winter to 65% in summer according to the GEOS-Chem vertical profiles. This needs to be accounted for when deriving NO$_x$ emissions from satellite NO$_2$ column measurements.

## 1 Introduction

Retrievals of NO$_2$ tropospheric columns from satellite measurements of solar backscatter are used extensively to infer anthropogenic NO$_x$ ($\equiv$NO+NO$_2$) emissions near the surface and their trends (e.g., Martin et al., 2003; Richter et al., 2005; Beirle et al., 2011; Krotkov et al., 2016). This is complicated by the presence of background NO$_2$ in the free troposphere, the part of the atmosphere between the top of the boundary layer (~2 km altitude) and the tropopause. NO$_x$ sources in the free troposphere include lightning, aircraft, transport from the boundary layer and the stratosphere, and chemical recycling from

HNO$_3$ and organic nitrates (Singh et al., 1996; Jaeglé et al., 1998a; Levy et al., 1999; Hudman et al., 2007). As fossil fuel NO$_x$ emissions have decreased in the US and other post-industrial countries, the relative contribution of the free tropospheric background to the tropospheric NO$_2$ columns has increased (Silvern et al., 2019). Satellite instruments are more sensitive to NO$_2$ in the free troposphere than in the boundary layer because of atmospheric scattering, so the NO$_2$ column retrievals must





assume a vertical distribution of $NO_2$ (shape factor) specified by an atmospheric chemistry model for the local conditions
(Martin et al., 2002; Eskes and Boersma, 2003). However, these models may be subject to large errors in the free troposphere
(Travis et al., 2016; Silvern et al., 2018). Here we use the vertical distribution of tropospheric $NO_x$ from aircraft measurements
over land and ocean, simulated with GEOS-Chem and other atmospheric chemistry models, to diagnose the confidence to be
had in these models and in the aircraft observations. We discuss the implications for global tropospheric oxidants and the
retrieval and interpretation of satellite $NO_2$ measurements in terms of surface $NO_x$ emissions.


Accurate in situ measurements of $NO_2$ in the free troposphere are challenging because of low $NO_2$ concentrations and
interferences from labile non-radical $NO_x$ reservoirs ($HNO_4$, $N_2O_5$, and organic nitrates) when sampling at cold temperatures
(Bradshaw et al., 1999; Browne et al., 2011; Reed et al., 2016; Nussbaumer et al., 2021). Current techniques to measure $NO_2$
*in situ* involve either (i) the conversion of $NO_2$ to NO by photolysis followed by measurement of NO through
chemiluminescence (photolysis-chemiluminescence; P-CL) (Walega et al., 1991; Ryerson et al., 2000; Bourgeois et al., 2022),
or (ii) the direct measurement of $NO_2$ through laser induced fluorescence (LIF) (e.g. Thornton et al., 2000; Matsumoto et al.,
2001; Javed et al., 2019), cavity ring-down spectroscopy (Osthoff et al., 2006), or cavity enhanced differential optical
absorption spectroscopy (Platt et al., 2009). Intercomparisons of $NO_2$ instruments have generally found agreement among the
different techniques at high (>1 ppbv) $NO_2$ concentrations (Thornton et al., 2003; Fuchs et al., 2010; Sparks et al., 2019;
Bourgeois et al., 2022), but poor agreement in free tropospheric conditions where $NO_2$ concentrations are below 50 pptv and
close to the instrument detection limits (Gregory et al., 1990a; Sparks et al., 2019). In contrast, NO measurements in the free
troposphere are generally found to be reliable down to about 10 pptv (Gregory et al., 1990a; Rollins et al., 2020). The $NO_2$
photolysis technique has been used for $NO_2$ measurements from aircraft since the 1980s (Ridley et al., 1988; Sandholm et al.,
1990). However, the free tropospheric $NO_2$ concentrations from these measurements were often found to be higher than
expected from $NO-NO_2$ photochemical steady state (PSS) (Davis et al., 1993; Fan et al., 1994; Crawford et al., 1996). This
was later attributed to an artifact in the $NO_2$ measurements from the thermal decomposition of peroxyacetyl nitrate (PAN),
$HNO_4$ and methyl peroxy nitrate (MPN) in the sample line and the photolysis cell (Bradshaw et al., 1999; Browne et al., 2011;
Reed et al., 2016). These species are present at relatively high concentrations at cold temperatures of the upper troposphere
(Murphy et al., 2004; Kim et al., 2007; Singh et al., 1986) and can cause significant interference in the $NO_2$ measurements
when the instrument temperature is higher than the ambient temperature (Nault et al., 2015; Reed et al., 2016).

The LIF technique was developed to eliminate interferences associated with the photolytic conversion of $NO_2$ (Thornton et al.,
2000) and has been widely used in aircraft campaigns to measure free tropospheric profiles of $NO_2$ over North America and
remote regions (Murphy et al., 2004; Bertram et al., 2007; Browne et al., 2011; Nault et al., 2015) and to evaluate satellite $NO_2$
retrievals (Bucsela et al., 2008; Boersma et al., 2008; Laughner et al., 2019). However, Silvern et al. (2018) found that the LIF
$NO_2$ measurements in the upper troposphere over the southeastern US during the Studies of Emissions and Atmospheric
Composition, Clouds and Climate Coupling by Regional Surveys (SEAC[4]RS) aircraft campaign were much higher than the




$NO_2$ concentrations expected from the NO-$NO_2$ PSS, indicating either an error in the NO-$NO_2$-$O_3$ kinetics at low temperatures or a remaining bias in the measurement.


Free tropospheric $NO_2$ concentrations have also been derived using remote sensing techniques. The Airborne Multi-AXis Differential Optical Absorption Spectroscopy (AMAX-DOAS) instrument can measure vertical profiles of $NO_2$ in the free troposphere (Baidar et al., 2013; Volkamer et al., 2015), but it is not routinely deployed to measure $NO_2$. Ground-based MAX-DOAS instruments can measure $NO_2$ vertical profiles in the boundary layer but have low sensitivity to the free troposphere (Vlemmix et al., 2011). $NO_2$ concentrations in the upper troposphere (8-12 km) have been retrieved from satellite $NO_2$ column measurements using cloud-slicing techniques based on measuring differences in partial $NO_2$ columns above clouds of different heights (Belmonte Rivas et al., 2015; Choi et al., 2014; Marais et al., 2021). These provide extensive spatial coverage but there are inconsistencies among different products and large differences compared to aircraft LIF measurements (Marais et al., 2018, 2021).


Atmospheric chemistry models are often used alongside satellite $NO_2$ measurements to determine surface $NO_x$ emissions and their trends, as they provide a way to relate changes in $NO_2$ columns to surface $NO_x$ emissions (Martin et al., 2003; Lamsal et al., 2011). But the sensitivity of modeled $NO_2$ columns to surface emissions depends on the relative contribution of the free troposphere to $NO_2$ columns. Modeled $NO_2$ vertical profiles over the continents generally agree with aircraft observations below about 6 km (Lamsal et al., 2014; Choi et al., 2020), but underestimate $NO_2$ measurements in the upper troposphere (Martin et al., 2006; Travis et al., 2016; Williams et al., 2017; Miyazaki et al., 2020). This could reflect model errors in the parametrized lightning $NO_x$ emissions (Martin et al., 2006; Allen et al., 2010; Hudman et al., 2007; Zhu et al., 2019), in convective transport of surface pollutants (Travis et al., 2016), or in $NO_x$ chemistry (Nault et al., 2016; Silvern et al., 2018). Silvern et al. (2018) showed that using the observed $NO_2$ vertical profile from SEAC[4]RS in the NASA $NO_2$ column retrieval for the OMI satellite instrument decreases the retrieved $NO_2$ columns over the southeastern US by 30%, suggesting the possibility of a systematic bias in the $NO_2$ column retrievals.

A number of global modeling studies have evaluated NO simulations over remote regions because of its importance for the production of tropospheric ozone and the hydroxyl radical (OH), and have generally found agreement within a factor of two (e.g., Emmons et al., 1997; Wang et al., 1998; Levy et al., 1999; Bey et al., 2001; Horowitz et al., 2003). However, a recent comparison of six global models with aircraft observations over the Pacific and Atlantic oceans made during the NASA Atmospheric Tomography (ATom) campaign's first deployment (July–August 2016) found significant underestimate of NO in all models below 4 km in the tropics and subtropics (Guo et al., 2021a). Other studies also suggest a missing source of $NO_x$ in models over the subtropical oceans from fast photolysis of particulate nitrate (Ye et al., 2016b; Reed et al., 2017; Kasibhatla et al., 2018; Andersen et al., 2022).



Here we use data from the SEAC4RS and the Deep Convective Clouds and Chemistry (DC3) aircraft campaigns to demonstrate the pervasiveness of interference from non-radical $NO_x$ reservoirs in $NO_2$ measurements in the upper troposphere. We go on to use the more reliable NO measurements and the $NO_2$ concentrations derived by applying PSS to the NO measurements to

evaluate the NO and $NO_2$ vertical profiles from different models for the SEAC4RS, DC3, and ATom campaigns. We use the model results to examine the sources of $NO_x$ in the free troposphere, effects on tropospheric ozone and OH, and background contribution to satellite $NO_2$ columns over the US.

## 2 Methods

### 2.1 Aircraft observations

We use observations from the SEAC4RS (August-September 2013; Toon et al., 2016) and DC3 (April-May 2012; Barth et al., 2015) campaigns over the southeastern US (25º–40ºN; 65º–100ºW), and the ATom campaign (4 seasonal deployments in 2016–18) over the Pacific and Atlantic oceans (Thompson et al., 2022). For all three campaigns, we use measurements from the NASA DC-8 aircraft, which has a ~12 km ceiling. Table 1 lists the measurements used in this work. Here, we briefly describe the $NO_2$ and NO measurements as they are most relevant. $NO_2$ measurements during the SEAC4RS and DC3

campaigns were made using the Berkeley LIF instrument (Thornton et al., 2000; Cleary et al., 2002; Nault et al., 2015). The LIF measurements have little (<5%) interference from $HNO_4$, but there is interference from decomposition of MPN, for which a correction was applied (0-21% for SEAC4RS and 0-40% for DC3) based on concurrent measurements of MPN from the same instrument (Nault et al., 2015). The LIF measurements have an accuracy of 5% and a detection limit of ~30 pptv for 1 Hz measurements (Thornton et al., 2000; Day et al., 2002; Wooldridge et al., 2010). $NO_2$ measurements in ATom were made

using the NOAA NOyO3 instrument using the P-CL technique (Ryerson et al., 2000; Bourgeois et al., 2022). The NOAA instrument also provided $NO_2$ measurements in SEAC4RS and DC3. The instrument has an accuracy of ~7% and a detection limit of 20–30 pptv for 1 Hz measurements (Pollack et al., 2010, 2012). Interference from dissociation of non-radical $NO_x$ reservoirs is lowered by reducing sample residence time and preventing heating of the photolysis cell (Pollack et al., 2010; Bourgeois et al., 2022). NO measurements in all three campaigns were made by the NOAA NOyO3 instrument, with an

accuracy of 4% and a detection limit of 6–10 pptv for 1 Hz measurements (Ryerson et al., 2000). For comparison with the model, we exclude measurements influenced by fresh convection (condensation nuclei larger than 10nm > $10^4$ $cm^{-3}$), fresh $NO_x$ emissions ($NO_y$/NO > 3 mol $mol^{-1}$), biomass burning plumes (CO > 200 ppbv and $CH_3CN$ > 200 pptv), and stratospheric intrusions ($O_3$ > 100 ppbv or CO < 45 ppbv).




**Table 1: Measurements from the SEAC[4]RS, DC3, and ATom aircraft campaigns[a]**

| Measurement | Instrument[b] | Campaigns | References |
|---|---|---|---|
| NO$_2$, MPN, alkyl nitrates | Berkeley TD-LIF | SEAC[4]RS, DC3 | Nault et al. (2015) |
| NO, NO$_2$, NO$_y$[c], O$_3$ | NOAA NOyO3 | SEAC[4]RS, DC3, ATom | Ryerson et al. (1998, 2000); Pollack et al. (2010); Bourgeois et al. (2020, 2022) |
| OH, HO$_2$ | Penn State ATHOS | DC3, ATom | Faloona et al. (2004); Brune et al. (2021) |
| HNO$_4$ | Georgia Tech CIMS | SEAC[4]RS, DC3 | Kim et al. (2007) |
| Photolysis frequencies | NCAR CAFS | SEAC[4]RS, DC3, ATom | Shetter and Müller (1999); Hall and Ullmann (2021) |
| Particulate nitrate | CU Boulder HR-AMS[d] | ATom | Hodzic et al. (2020); Nault et al. (2021) |
| | UNH SAGA[d] | | Dibb et al. (1999); Dibb (2020) |
| Condensation nuclei | NASA Langley CPC (TSI 3772) | SEAC[4]RS, DC3 | [e] |
| CO | NASA Langley DACOM | SEAC[4]RS, DC3 | Sachse et al. (1991) |
| | NOAA Picarro (G2401) | ATom | Chen et al. (2013); McKain and Sweeney (2021) |
| CH$_3$CN | Innsbruck PTR-MS | SEAC[4]RS, DC3 | Wisthaler et al. (2002) |

[a] Measurements used in this work to evaluate the NO$_x$ simulations and to select data for analysis
[b] Instrument acronyms: TD-LIF: Thermal Dissociation, Laser Induced Fluorescence; ATHOS: Airborne Tropospheric Hydrogen Oxides Sensor; CIMS: Chemical Ionization Mass Spectrometer; CAFS: CCD Actinic Flux Spectroradiometer; HR-AMS: High Resolution Aerosol Mass Spectrometer; SAGA: Soluble Acidic Gases and Aerosols; CPC: Condensation Particle Counter; DACOM: Differential Absorption Carbon mOnoxide Monitor; PTS-MS: Proton Transfer Reaction - Mass Spectrometry
[c] Total reactive nitrogen oxides including NO$_x$ and its oxidation products
[d] The AMS measures the composition of non-refractory submicron aerosols. SAGA measures the ionic composition of water-soluble bulk aerosols of diameter less than about 4 μm.
[e] Commercial instrument operated by the NASA Langley Aerosol Research Group Experiment

**2.2 GEOS-Chem model**

We use the GEOS-Chem atmospheric chemistry model (12.9.3; doi: 10.5281/zenodo.3959279), with modification to include particulate nitrate (pNO$_3$$^-$) photolysis as described below. Our simulations are driven by assimilated meteorology from NASA GMAO's Modern-Era Retrospective analysis for Research and Applications, Version 2 (MERRA-2; Gelaro et al., 2017). We conduct global simulations at 4º✕5º horizontal resolution (47 levels in the vertical) for the time periods corresponding to the aircraft campaigns: SEAC[4]RS (July–August 2013), DC3 (May–June 2012), ATom (July–August 2016, January–February

2017, September–October 2017, and April–May 2018), as well as an annual simulation for 2015. Previous work on the SEAC[4]RS campaign used finer resolution simulations (Travis et al., 2016), but these are not needed here as the free tropospheric NO$_2$ concentrations do not vary much at regional scales and finer resolution tests showed similarity in results (Yu



et al., 2016). The horizontal grid resolution can lead to localized differences in the upper troposphere from stratospheric intrusions, convective transport, and lightning $NO_x$ emissions (Schwantes et al., 2022), and we minimize these effects by filtering out data influenced by the stratosphere, fresh convection, and fresh $NO_x$ emissions, as described above. The spin-up period for our simulations is six months. Comparison to aircraft measurements is done by sampling the model along the flight path.

Emissions in GEOS-Chem are calculated by the Harmonized Emissions Component (HEMCO) (Keller et al., 2014) with updated inventories. Table 2 lists the global $NO_x$ emissions in our 2015 simulation. Anthropogenic $NO_x$ emissions are from the Community Emissions Data System (CEDS) global inventory (Hoesly et al., 2018), superseded with regional emission inventories for the US (US EPA 2011 NEI, 2016), Canada (Air Pollutant Emissions Inventory, 2017), Africa (Marais and Wiedinmyer, 2016), and China (Zheng et al., 2018). The US EPA 2011 NEI is scaled annually using EPA-estimated emissions trends (US EPA Air Pollutant Emissions Trends Data, 2015). Travis et al. (2016) had to scale down the NEI $NO_x$ emissions in GEOS-Chem by 40% to reproduce the SEAC$^4$RS $NO_x$ observations, but we do not do this in our simulations as it leads to an underestimate in $NO_x$ in other seasons (Jaeglé et al., 2018; Silvern et al., 2019). Open fire $NO_x$ emissions are from the GFEDv4 inventory (Giglio et al., 2013). Ship $NO_x$ emissions are from CEDS and are processed using the PARAmetrization of emitted NOX (PARANOX) model to account for fast in-plume $NO_x$ oxidation (Vinken et al., 2011; Holmes et al., 2014). Aircraft $NO_x$ emissions are from the Aviation Emissions Inventory Code (AEIC) inventory (Stettler et al., 2011; Simone et al., 2013), and are updated here with flight traffic data for 2015. Lightning $NO_x$ emissions follow Murray et al. (2012), with lightning flash rates calculated as a function of the cloud top height and scaled to match the observed climatology from satellite data. Emissions are computed at the native MERRA-2 resolution (0.5º×0.625º). NO yields of 500 moles per flash are used for the northern midlatitudes (>35ºN) and 260 moles per flash elsewhere. Emissions are distributed in the vertical following Ott et al. (2010). Soil and fertilizer $NO_x$ emissions are from Hudman et al. (2012) and are computed at 0.5º×0.625º resolution (Weng et al., 2020).

**Table 2: Global NO$_x$ emissions in 2015[a]**

| Source[b] | Emission rate (TgN a$^{-1}$) |
|---|---|
| Fuel combustion | 35.2 |
| Fires | 6.6 |
| Soils & fertilizer use | 8.1 |
| Aircraft | 1.2 |
| Lightning | 5.8 |
| Total[c] | 56.9 |

[a] as used in our GEOS-Chem simulation
[b] references for the different sources are given in the text
[c] not including the $NO_x$ source of ~0.5 TgN a$^{-1}$ from downwelling of stratospheric $NO_y$ produced from $N_2O$





GEOS-Chem includes a detailed representation of $NO_x$-$HO_x$-VOC-aerosol-halogen chemistry (Mao et al., 2013; Travis et al., 2016; Holmes et al., 2019; Wang et al., 2021; McDuffie et al., 2021; Pai et al., 2020). Recent improvements to the model's

$NO_x$ chemistry include addition of detailed tropospheric halogen chemistry (Wang et al., 2021), addition of methyl, ethyl, and propyl nitrate emissions and chemistry (Fisher et al., 2018), and updates to the heterogeneous $NO_x$ reactions in aerosols and cloud droplets (Holmes et al., 2019; McDuffie et al., 2021). Here we follow Schmidt et al. (2016) and exclude bromine release from sea salt aerosol debromination because it leads to excessive model BrO in the marine boundary layer (MBL). Equilibrium partitioning of $HNO_3$ to $pNO_3^-$ on fine mode aerosols is calculated using ISORROPIA II (Fountoukis and Nenes, 2007; Wang

et al., 2019). Uptake of $HNO_3$ as $pNO_3^-$ on coarse sea salt aerosols is treated as a kinetic process, following Wang et al. (2019). Sea salt aerosol emissions follow Jaeglé et al. (2011) and are calculated at $0.5°×0.625°$ resolution (Weng et al., 2020). Our simulation does not include $HNO_3$ uptake on alkaline dust particles, but this could be important in dust plumes over the ocean (Fairlie et al., 2010; Karydis et al., 2016). Photolysis frequencies in the model are calculated using Fast-JX (Wild and Prather, 2000; Eastham et al., 2014).


Previous studies examining the GEOS-Chem NO simulation for the ATom campaign showed underestimates in the lower troposphere (Fisher et al., 2018; Travis et al., 2020; Guo et al., 2021a). Measurements in the marine atmosphere indicate elevated levels of HONO that originate likely from $pNO_3^-$ photolysis (Ye et al., 2016b; Andersen et al., 2022) and would provide a fast source of $NO_x$ missing from the model. We address this by including $pNO_3^-$ photolysis in our simulation,

following the implementation of this reaction in GEOS-Chem by Kasibhatla et al. (2018). The photolysis frequency of $pNO_3^-$ is calculated by scaling the photolysis frequency of $HNO_3$ by an enhancement factor (EF). There is high uncertainty in the EF, with laboratory studies in the range of 1–1000 (Ye et al., 2016a; Bao et al., 2018; Gen et al., 2019; Shi et al., 2021). Field and modeling studies find that EFs of 10–500 are needed to explain the $NO_x$ and HONO observations over the oceans (Ye et al., 2016b, 2017a; Reed et al., 2017; Kasibhatla et al., 2018; Zhu et al., 2022; Andersen et al., 2022), with higher values for $pNO_3^-$

in sea salt aerosols (Andersen et al., 2022). In consistency with these studies, we find that we can match the ATom NO observations using an EF of 100 for $pNO_3^-$ in sea salt aerosol. In our model, coarse mode $pNO_3^-$ is only present in sea salt aerosols and has an EF of 100, but fine mode $pNO_3^-$ is internally mixed with sulfate, ammonium, and sea salt aerosol and so we decrease the EF of fine mode $pNO_3^-$ depending on the relative amounts of $pNO_3^-$ and sea salt aerosol:

$$EF = 100 \times \frac{1}{1+\frac{[pNO_3^-]}{[SSA]}}, \quad EF_{min}=10 \tag{1}$$

Here $[pNO_3^-]$ and [SSA] are the molar concentrations in air of fine mode $pNO_3^-$ and sea salt aerosol. The molar concentration of sea salt is taken as $[SSA] = 2.39[Na^+]$ based on seawater salt composition, and where $Na^+$ is the chemically inert sea salt aerosol species simulated by GEOS-Chem. We choose a lower limit for the EF ($EF_{min}$) of 10 based on the results of Romer et al. (2018), who estimated EF values for non-sea-salt $pNO_3^-$ aerosols of 1–30 from observations over South Korea. The relative



yields of HONO:NO$_2$ from pNO$_3^-$ photolysis are taken as 2:1 (Ye et al., 2017b; Kasibhatla et al., 2018). We will discuss the effect of pNO$_3^-$ photolysis on background NO$_x$ in more detail in Sections 3.2.

## 2.3 Other models

In addition to GEOS-Chem simulations, we analyze results from three other global atmospheric chemistry models: the Global
Modeling Initiative (GMI) model, the Tracer Model version 5's "massively parallel" version (TM5-MP), and the Copernicus Atmosphere Monitoring Service (CAMS) reanalysis product. The GMI model simulates tropospheric and stratospheric chemistry (Duncan et al., 2007; Strahan et al., 2007; Strode et al., 2015) and uses meteorological fields from NASA GMAO's MERRA-2 reanalysis. GMI NO$_2$ vertical profiles are used in the OMI NO$_2$ retrievals (Krotkov et al., 2017; Lamsal et al., 2021). The version used here has a horizontal resolution of 1°×1.25°. Strode et al. (2021) describe the GMI model simulations
for the ATom campaign. The TM5-MP model is a high resolution (1°×1°) version of the TM5 global atmospheric chemistry model developed specifically for application to satellite retrievals (Williams et al., 2017; Huijnen et al., 2010). It is driven by assimilated meteorology from the European Centre for Medium-Range Weather Forecasts (ECMWF) ERA-Interim reanalysis. The TM5 NO$_2$ profiles are used in the Quality Assurance for Essential Climate Variables (QA4ECV) OMI and TROPOMI NO$_2$ retrievals (Boersma et al., 2018; van Geffen et al., 2022). CAMS provides a global reanalysis of atmospheric composition
at a horizontal resolution of 80 km (T255) for the period 2003 onwards (Inness et al., 2019). It is based on ECMWF's Integrated Forecast System (IFS) and uses 4D-Var data assimilation of satellite retrievals of NO$_2$, O$_3$, CO, and aerosol optical depth. The CAMS NO$_2$ profiles are planned for use in NO$_2$ retrievals from the European Sentinel-4 geostationary satellite (ESA Sentinel-4 Data Products, 2022). The TM5 and CAMS output along the ATom flight tracks was available only for the first ATom deployment (July–Aug 2016).

## 3 Results and discussion

### 3.1 Vertical distribution of NO$_x$ over the US

Figure 1 compares the median vertical profiles of the observed and GEOS-Chem NO and NO$_2$ concentrations for the SEAC$^4$RS and DC3 aircraft campaigns. For both campaigns, the observed and GEOS-Chem NO concentrations peak in the boundary layer, and again in the upper troposphere because of lightning and aircraft emissions, convective lifting of surface emissions,
long NO$_x$ lifetime (except near fresh convection), and shift of the daytime NO/NO$_2$ ratio toward NO at low temperatures (Jaeglé et al., 1998a; Bertram et al., 2007; Hudman et al., 2007; Nault et al., 2017). The GEOS-Chem NO$_2$ profiles are similar to the LIF NO$_2$ profiles below 10 km but differ in the upper troposphere, as previously noted by Travis et al. (2016). LIF NO$_2$ concentrations in SEAC$^4$RS increase from 20 pptv at 9 km to 120 pptv at 12 km, but GEOS-Chem NO$_2$ concentrations remain below 30 pptv. The difference between GEOS-Chem and the P-CL NO$_2$ observations in the upper troposphere during DC3 is
even larger.





**Figure 1. Median vertical profiles of observed and GEOS-Chem simulated NO and NO₂ concentrations, and NO/NO₂ molar ratios, during the SEAC⁴RS (Aug–Sep 2013) and DC3 (Apr–May 2012) aircraft campaigns over the southeastern US. The NO**
**measurements are from the NOAA P-CL instrument. NO₂ was measured by the Berkeley LIF and the NOAA P-CL instruments. The NO/NO₂ ratios at photochemical steady state (PSS; Eq. 2) and the corresponding NO₂ concentrations (Eq. 3) are also shown in the rightmost panels. Also shown are the NO/NO₂* (NO₂*≡NO₂+HNO₄+MPN) ratios from GEOS-Chem. MPN is methyl peroxy nitrate. We exclude measurements in early mornings and late evenings (solar zenith angle >70º) and influenced by fresh NOₓ emissions recent convection, biomass burning, and the stratosphere as described in Sect. 2.1. The horizontal bars show the**
**interquartile ranges of the measurements in each 1-km altitude bin.**



Travis et al. (2016) and Silvern et al. (2018) showed that the difference between the measured and GEOS-Chem $NO_2$ in the upper troposphere in $SEAC^4RS$ can be explained by the departure of the measured $NO/NO_2$ ratio from that expected from calculated PSS between NO and $NO_2$. In daytime, NO and $NO_2$ interconvert rapidly through the following main reactions:


$$NO + O_3 \longrightarrow NO_2 + O_2 \tag{R1}$$

$$NO + HO_2 \longrightarrow NO_2 + OH \tag{R2}$$

$$NO + RO_2 \longrightarrow NO_2 + RO \tag{R3}$$

$$NO + BrO \longrightarrow NO_2 + Br \tag{R4}$$

$$NO_2 + h\nu \xrightarrow{+O_2} NO + O_3 \tag{R5}$$

Here $RO_2$ represents the ensemble of organic peroxy radicals. At PSS, the $NO/NO_2$ ratio is given by:

$$PSS = \frac{[NO]}{[NO_2]} = \frac{j_{NO_2}}{k_1[O_3] + k_2[HO_2] + k_3[RO_2] + k_4[BrO]} \tag{2}$$

where $j_{NO_2}$ is the $NO_2$ photolysis frequency and $k_i$ is the rate constant of reaction $i$. We calculate the PSS $NO/NO_2$ ratio for the $SEAC^4RS$ and DC3 data using concurrent aircraft measurements and GEOS-Chem simulated values along the flight path for quantities that were not measured. $[O_3]$ and $j_{NO_2}$ were measured in both campaigns. $[HO_2]$ was measured only in DC3, but

$H_2O_2$ concentrations measured in $SEAC^4RS$ are consistent with GEOS-Chem (Silvern et al., 2018), which provides support for the model $HO_2$ values. Rate constants are as recommended by the JPL evaluation (Burkholder et al., 2020) and adjusted for temperature and pressure. We assume $k_3$ equals $k_2$. $RO_2$ and BrO concentrations are taken from GEOS-Chem but make only small contributions. In the free troposphere, $NO-NO_2$ PSS is largely governed by the $NO + O_3$ reaction (Bradshaw et al., 1999; Silvern et al., 2018). Thus, the PSS $NO/NO_2$ ratio depends mainly on observed quantities and on relatively well-

established kinetics (Silvern et al., 2018). We estimate the uncertainty in the PSS $NO/NO_2$ ratio at 1 Hz of about ±20%, based on uncertainties in $[O_3]$ (0.015 ppbv ± 2%; Bourgeois et al., 2020), $[HO_2]$ (35%; Brune et al., 2021), $j_{NO_2}$(15%; Hall and Ullmann, 2021), rate constants (10%; Burkholder et al., 2020), and model $[RO_2]$ and [BrO] (assumed to be 50%).

Figure 1 compares the vertical profiles of the measured and the PSS $NO/NO_2$ ratios. The PSS $NO/NO_2$ ratio increases with

altitude because of the slower rate of the $NO + O_3$ reaction at colder temperatures (Burkholder et al., 2020). The measured and PSS ratios match below 5 km, but at higher altitudes the measured ratios are smaller than the PSS ratios. Between 10 and 12 km, the $NO/NO_2$ ratios using the LIF measurements are in the range of 1 to 2, while the PSS $NO/NO_2$ ratios are in the range of 3 to 6. The $NO/NO_2$ ratios using the P-CL measurements at this altitude are close to 1. The GEOS-Chem $NO/NO_2$ ratios are similar to the PSS ratios throughout the troposphere.


The P-CL $NO_2$ instrument is known to have interference from dissociation of $HNO_4$ and MPN (Reed et al., 2016; Nussbaumer et al., 2021), but the magnitude of the interference has not been quantified. We find that the ratio of $NO/NO_2^*$ ($NO_2^* \equiv NO_2 + HNO_4 + MPN$) in GEOS-Chem closely matches the $NO/NO_2$ ratio for the P-CL $NO_2$ measurements, suggesting



that HNO$_4$ and MPN dissociate in the instrument with the same efficiency as that of NO$_2$. The LIF NO$_2$ measurements correct

for such interferences, but the correction is affected by the high uncertainty in the concentrations and the thermal stability of

MPN (Nault et al., 2015). The LIF instrument was modified between DC3 and SEAC$^4$RS to shorten the sample residence time

and reduce the fraction of MPN dissociating in the instrument (Nault et al., 2015), but we do not find that this improved

agreement between the measured and PSS NO/NO$_2$ (Fig. 1).

Silvern et al. (2018) hypothesized that the difference between the measured and PSS NO/NO$_2$ ratios could arise from either an

error in the NO-NO$_2$-O$_3$ kinetics or a systematic bias in the NO$_2$ measurements in the upper troposphere. Here we arbitrate

between these two hypotheses by using quasi-Lagrangian observations in a thunderstorm outflow in the upper troposphere

deliberately sampled during DC3 (flight RF17). Nault et al. (2016) previously analyzed the evolution of NO$_x$ and NO$_y$ (NO$_y$ ≡

NO$_x$ + non-radical reservoirs) on this flight to determine NO$_x$ oxidation rates. Figure 2 shows the flight path with daytime

plume crossings colored by the measured NO$_y$/NO molar ratio. The NO$_y$/NO ratio increases on each successive plume crossing

as the lightning-generated NO$_x$ in the thunderstorm undergoes oxidation in the outflow, and we use the ratio as a measure of

chemical aging in the plume (Kleinman et al., 2008; Hayes et al., 2013). Also shown in Fig. 2 are the measured NO, NO$_2$, and

the sum of HNO$_4$ and MPN concentrations as a function of the NO$_y$/NO ratio. NO concentrations decreased from 900 pptv to

400 pptv between the start and the end of the measurement period. In comparison, the LIF NO$_2$ concentrations decreased by

25%, while the P-CL NO$_2$ concentrations increased, likely due to increasing interference from HNO$_4$ and MPN produced in

the plume. Figure 2 also show the NO$_2$ concentrations inferred by applying PSS to NO observations:

$$[NO_2]_{PSS} = \frac{[NO]}{PSS} , \tag{3}$$

where PSS is calculated from observations using Eq. (2) and measured [O$_3$], [HO$_2$] and $j_{NO_2}$. In this case, we take [RO$_2$] to be

equal to the measured [HO$_2$], instead of using the value from GEOS-Chem, since we do not expect the model to simulate the

thunderstorm plume. The PSS NO$_2$ concentrations decreased by a factor of 2 between the start and end of the measurement

period, in line with the NO concentrations.

The bottom panel shows the observed ozone concentrations as a function of the NO$_y$/NO ratio. Ozone concentrations increase

along the plume, reflecting the NO$_x$-limited conditions for ozone production prevalent in the upper troposphere over the central

US (Pickering et al., 1990; Jaeglé et al., 1998b; Apel et al., 2015). We compare the observed ozone increase to that computed

from the observed $j_{NO_2}$ and the observed NO, NO$_2$, HO$_2$, and OH concentrations. Ozone is produced through the photolysis of

NO$_2$ (reaction R5), and is lost mainly by reaction with NO (reaction R1), photolysis in the presence of water vapor (reaction

R6), and oxidation by HO$_2$ and OH (reactions R7 and R8):

$$O_3 + h\nu \xrightarrow{+H_2O} 2OH + O_2 \tag{R6}$$

$$O_3 + HO_2 \longrightarrow 2O_2 + OH \tag{R7}$$
$$O_3 + OH \longrightarrow HO_2 + O_2 \tag{R8}$$





The instantaneous net ozone production rate is then given as follows:

$$\frac{d\,[\mathrm{O_3}]}{dt} = j_{\mathrm{NO_2}}[\mathrm{NO_2}] - k_{\mathrm{NO+O_3}}[\mathrm{NO}][\mathrm{O_3}] - k_{\mathrm{O_3 \to OH}}[\mathrm{O_3}] - k_{\mathrm{HO_2+O_3}}[\mathrm{HO_2}][\mathrm{O_3}] - k_{\mathrm{OH+O_3}}[\mathrm{OH}][\mathrm{O_3}] \qquad (4)$$

We use Eq. (4) to calculate three estimates for the instantaneous net ozone production rate in the plume using $\mathrm{NO_2}$ from LIF,

P-CL, and PSS. The total ozone increase in the plume is calculated by integrating $\frac{d\,[\mathrm{O_3}]}{dt}$ over the measurement period.

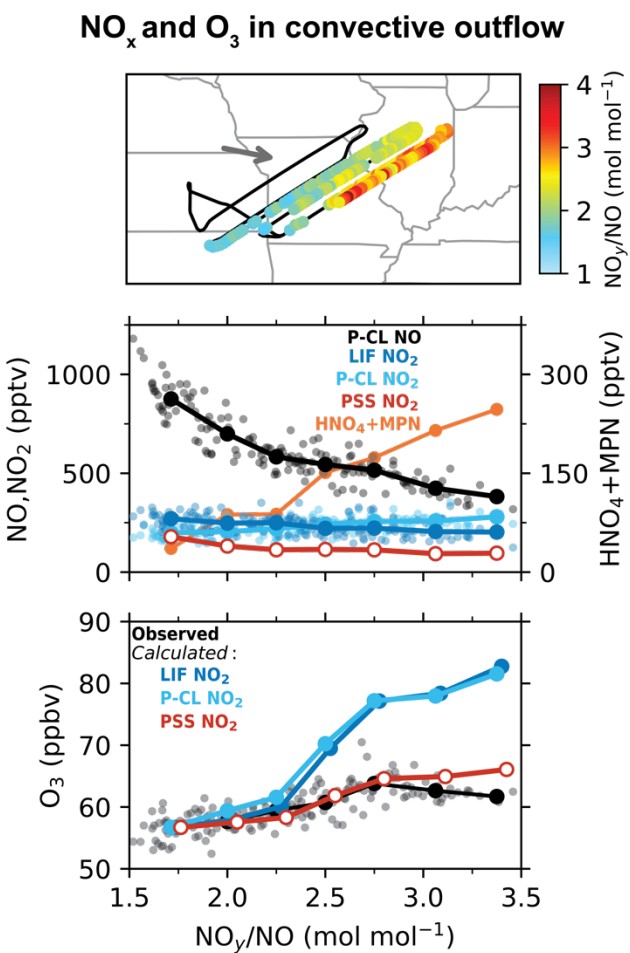

**Figure 2. Evolution of $\mathrm{NO_x}$ and $\mathrm{O_3}$ concentrations in a thunderstorm outflow targeted for quasi-Lagrangian sampling by DC3 flight 17 at 12 km altitude. The top panel shows the flight track with data points within the outflow colored by the observed $\mathrm{NO_y/NO}$ ratio**
**as a measure of chemical aging. The arrow shows the mean wind direction. The lower two panels show the measured concentrations of NO, $\mathrm{NO_2}$, ozone, and the sum of $\mathrm{HNO_4}$ and MPN as a function of the $\mathrm{NO_y/NO}$ molar ratio. Also shown are the PSS $\mathrm{NO_2}$ concentrations (Eq. 3) and the evolution of ozone concentrations calculated from Eq. (4) with $\mathrm{NO_2}$ concentrations from LIF, P-CL, or PSS $\mathrm{NO_2}$. Symbols show individual 1-minute observations and solid lines with circles show median values for $\mathrm{NO_y/NO}$ bins.**

The observed ozone concentrations increased by 7 ppbv between the start and the end of the measurement period in the plume.

In comparison, the ozone increase calculated using the $\mathrm{NO_2}$ measurements from both the LIF and P-CL instruments is 25 ppbv,



but that calculated using the PSS $NO_2$ concentrations is close to the observations. We also examine the effect of potential uncertainties in the $NO$-$NO_2$-$O_3$ kinetic data by decreasing $j_{NO_2}$ by 20% and increasing $k_{NO+O_3}$ by 40% in Eq. (4), following Silvern et al. (2018). We find that the ozone increase calculated using the $NO_2$ measurements is lowered to 17 ppbv, still much

higher than the observed increase, and implying that the difference between the $NO_2$ measurements and the PSS $NO_2$ concentrations cannot be attributed to errors in the $NO$-$NO_2$-$O_3$ kinetic data. The most likely explanation is that the LIF $NO_2$ measurements are also biased high, as are the P-CL measurements. The median LIF and P-CL $NO_2$ concentrations in the outflow plume were both 235 pptv, compared to a median PSS $NO_2$ concentration of 116 pptv. The median measured $HNO_4$ and MPN concentrations were 44 and 90 pptv, respectively, and can explain the difference between the P-CL and PSS $NO_2$

concentrations. The LIF $NO_2$ measurements are thought to have little interference from $HNO_4$, and were corrected for the partial dissociation of MPN, but it appears that this correction may have been underestimated. For this flight, the median correction to the $NO_2$ measurements was just 7%. The correction is affected by high uncertainty in the thermal dissociation rate constant of MPN (±30%) and in the MPN measurements (±40%+20 pptv for 1 Hz; (Nault et al., 2015). The MPN measurements themselves would be affected by a bias in the $NO_2$ measurements as they are based on the difference in the $NO_2$

measured between a heated channel that dissociates MPN and the unheated $NO_2$ channel. Interference from other known non-acyl peroxy nitrates would not be significant (Khan et al., 2020), but there could be other unknown organic $NO_2$ reservoir species that could form in convective outflows (Silvern et al., 2018).

Considering this bias in the LIF and P-CL $NO_2$ measurements in the upper troposphere, we instead use the $NO$ observations

and the PSS $NO_2$ concentrations inferred from the $NO$ and other observations (Eq. 3) to evaluate the modeled $NO_x$ in the free troposphere (Fig. 1). GEOS-Chem reproduces the shape of the $NO$ and the PSS $NO_2$ profiles throughout the troposphere for SEAC[4]RS and DC3. There is no increase in the modeled or the PSS $NO_2$ concentrations in the upper troposphere, as higher $NO$ concentrations are compensated by higher $NO/NO_2$ ratios. GEOS-Chem $NO$ concentrations are about 2 times higher than the observations in the free troposphere, consistent with previous work for SEAC[4]RS (Travis et al., 2016; Silvern et al., 2018).

The PSS $NO_2$ column density in the free troposphere for SEAC[4]RS and DC3 is $3.6 \times 10^{14}$ and $3.8 \times 10^{14}$ molec cm$^{-2}$, respectively, compared to $6.5 \times 10^{14}$ and $10.4 \times 10^{14}$ molec cm$^{-2}$ in GEOS-Chem. However, the model does not overestimate $NO_y$ concentrations, suggesting that the model may be missing $NO_x$ oxidation chemistry, which is likely organic. We find that the median MPN concentration in the free troposphere in GEOS-Chem is about 5 pptv compared to about 40 pptv in the observations, consistent with the findings of Silvern et al. (2018) for SEAC[4]RS. Similarly, median alkyl nitrate concentration

in the model is about 12 pptv but 60 pptv in the observations. $NO_x$ emissions are likely overestimated in the US EPA NEI inventory used in our simulations (Travis et al., 2016), which explains the $NO_2$ overestimate in the boundary layer, but this would have little effect in the free troposphere, where lightning emissions supply the majority of $NO_x$. Finally, we find little difference in the SEAC[4]RS and DC3 $NO_x$ profiles in the free troposphere between our baseline simulation and the simulation without $pNO_3^-$ photolysis, indicating that chemical recycling through $pNO_3^-$ photolysis is a minor source of $NO_x$ over the US

compared to emissions.



Retrieval of $NO_2$ columns from satellite-based instruments generally involves the following steps: (i) using the observed solar backscatter radiance to calculate a total slant $NO_2$ column density along the light path, (ii) removal of the stratospheric contribution to calculate the tropospheric slant column density $\Omega_s$, and (iii) conversion of the tropospheric slant column density

to a tropospheric vertical column density $\Omega_v$, using an air mass factor (AMF) that depends on the vertical profile of $NO_2$ (Palmer et al., 2001; Martin et al., 2002):

$$\frac{\Omega_s}{\Omega_v} = AMF = AMF_G \int_0^{z_t} w(z)\, S(z)\, dz, \tag{5}$$

where $AMF_G$ is the geometric AMF that describes the satellite viewing geometry, $w(z)$ are the scattering weights that describe the sensitivity of the backscattered radiance to the $NO_2$ abundance as a function of altitude ($z$), $S(z)$ is the $NO_2$ shape factor

describing the vertical profile of the $NO_2$ number density normalized to the $NO_2$ vertical column density, and $z_t$ is the tropopause height. $w(z)$ is computed with radiative transfer modeling, and in clear skies is 3–4 times higher in the upper troposphere than in the boundary layer because of atmospheric scattering (Martin et al., 2002). Here we use scattering weights from the NASA OMI $NO_2$ retrieval (v4.0; Lamsal et al., 2021), and exclude scenes with clouds (cloud fraction > 0.1) and bright surfaces (surface albedo > 0.3).


We use Eq. 5 to calculate AMFs corresponding to PSS and GEOS-Chem $NO_2$ profiles for SEAC⁴RS and DC3. $AMF_G$ over the southeastern US in summer for OMI is about 2.6. For SEAC⁴RS, both the PSS and GEOS-Chem $NO_2$ profiles yield an AMF of 1.0, reflecting the similar shapes of the $NO_2$ profiles, although GEOS-Chem overestimates the absolute values. For DC3, the AMFs corresponding to the PSS and GEOS-Chem profiles are 0.91 and 1.03, respectively. These results suggest that

using the GEOS-Chem $NO_2$ profiles as *a priori* in the $NO_2$ column retrievals over the southeast US would result in an error of 0–10%, compared to the previous error estimate of 30% based on the LIF $NO_2$ measurements in SEAC⁴RS (Silvern et al., 2018). The sensitivity of satellite retrievals to $NO_2$ vertical profiles is discussed further in Section 3.5.

### 3.2 $NO_x$ in the remote troposphere: interpreting the ATom data

We now examine the distribution of $NO_x$ over the Pacific and Atlantic oceans during the ATom campaign in order to

characterize the global background $NO_2$ and contrast it to the SEAC⁴RS and DC3 $NO_2$ profiles over land. Modeled $NO_2$ over remote regions is often used in the stratospheric-tropospheric separation of satellite $NO_2$ columns (Bucsela et al., 2013). In addition, $NO_x$ in the remote troposphere is important for global tropospheric ozone and OH production. Figures 3 and 4 show the median vertical profiles of NO and the PSS $NO_2$ concentrations over the Pacific and Atlantic Oceans separated by seasons and latitude bands. The PSS $NO_2$ concentrations in Fig. 4 are inferred from the ATom observations of NO, ozone, $HO_2$ and

$j_{NO_2}$ using Eqs. (2) and (3). The observed NO concentrations increase from 10 pptv near the surface to 20–100 pptv in the upper troposphere above 8 km because of the longer $NO_x$ lifetime and the increase in $NO/NO_2$ ratios with altitude. The PSS $NO_2$ profiles show a decrease in $NO_2$ concentrations with altitude because of an increase in the $NO/NO_2$ ratio. PSS $NO_2$





concentrations in the upper troposphere are generally lower than 10 pptv, except in the northern midlatitudes upper troposphere
in August and October, where $NO_2$ concentrations increase in the upper troposphere. The upper tropospheric $NO_x$

concentrations over the Atlantic in August are similar to those observed over the southeastern US during SEAC[4]RS and DC3
and reflect the transport of lightning-generated $NO_x$ from the US to the Atlantic Ocean (Crawford et al., 2000; Cooper et al.,
2006; Singh et al., 2007). There is little seasonal variation in $NO_x$ below 8 km. The column density for PSS $NO_2$ has a campaign
mean of $1.9 \times 10^{14}$ molec cm[-2] and a range of $1.2$–$3.0 \times 10^{14}$ molec cm[-2] for the different seasons and latitude bands. The free
tropospheric PSS $NO_2$ column density over the northern Atlantic (30–60ºN) in August is $2.1 \times 10^{14}$ molec cm[-2], about 45%

lower than that during SEAC[4]RS and DC3.



**Figure 3. Median vertical profiles of NO concentrations over the Pacific and Atlantic Oceans during the ATom flight campaigns**
**(2016–18), separated by seasons and latitude bands. Observations (black) are from the NOAA P-CL instrument. The data selection**
**criteria are as described in the caption of Fig. 2. Horizontal bars show the interquartile ranges in 1-km altitude bins. Model results**
**are from our baseline GEOS-Chem simulation and a sensitivity simulation without pNO3- photolysis. The model is sampled along**
**the flight tracks. NO concentrations are plotted on a log scale.**






**Figure 4. Median vertical profiles of NO$_2$ concentrations over the Pacific and Atlantic Oceans during the ATom flight campaigns (2016–18), separated by seasons and latitude regions. Observations are based on photochemical steady state (PSS) with local measurements of NO concentrations and other quantities following Eqs. (2) and (3). Horizontal bars show the interquartile ranges in 1-km altitude bins. The data selection criteria are as described in the caption of Fig. 2. NO$_2$ measurements from the P-CL instrument are also shown for reference. Model results are from our baseline GEOS-Chem simulation (including pNO$_3^-$ photolysis), GMI, TM5, and CAMS, sampled along the flight tracks. The TM5 and CAMS NO$_2$ profiles are available only for August. NO$_2$ concentrations are plotted on a log scale.**

Figure 3 compares the NO observations to results from our baseline GEOS-Chem simulation and from a sensitivity simulation without the NO$_x$ source from pNO$_3^-$ photolysis. The GEOS-Chem simulation without pNO$_3^-$ photolysis underestimates NO observations below 6 km by a factor of 2–5 in most cases. The underestimate does not extend to the upper troposphere so it cannot be attributed to errors in lightning or aircraft NO$_x$ emissions. GEOS-Chem does not underestimate ATom HNO$_3$ (Travis





et al., 2020; Luo et al., 2020) or PAN (Zhai et al., 2022) observations, so the underestimate in NO is not related to $NO_x$ recycling from these species. There is no sign of a significant overestimate in the $NO_x$ sinks in the model either. GEOS-Chem's

OH concentrations are consistent with ATom observations (Travis et al., 2020). There is some uncertainty in the $NO_2+OH+M\rightarrow HNO_3+M$ rate constant used in models (Mollner et al., 2010; Nault et al., 2016; Burkholder et al., 2020), but not large enough to explain the NO underestimate. The model's heterogenous $NO_x$ chemistry reflects current knowledge and includes an empirical parameterization for the $N_2O_5$ reaction probability derived from the aircraft observations (Jaeglé et al., 2018; McDuffie et al., 2018; Holmes et al., 2019).


Recent studies suggest that photolysis rate of $pNO_3^-$ could be much faster than the photolysis of gas-phase $HNO_3$, which could make this an important source of $NO_x$ over the oceans (Ye et al., 2016a, b; Reed et al., 2017; Kasibhatla et al., 2018). $pNO_3^-$ photolysis produces $NO_2$ and HONO (Scharko et al., 2014; Ye et al., 2017b), and HONO photolyzes further to produce NO:

$$pNO_3^- + h\nu \xrightarrow{\text{H}_2\text{O (l)}} HONO(g) + OH^- + O(^3P) \tag{R9a}$$


$$\xrightarrow{\text{H}_2\text{O (l)}} NO_2(g) + OH^- + OH \tag{R9b}$$

$$HONO + h\nu \longrightarrow NO + OH \tag{R10}$$

In bulk solution, the absorption cross-section of $NO_3^-$ is about 100 times larger than that of $HNO_3$ (Burley and Johnston, 1992) but the effective quantum yields for Reactions (R9a) and (R9b) are low (~1%) (Warneck and Wurzinger, 1988; Benedict et

al., 2017), because products are surrounded by water molecules and recombine before they can escape to the gas phase (Nissenson et al., 2010; Richards-Henderson et al., 2015). However, the photolysis of $NO_3^-$ on aerosols is thought to be much more efficient than that in the gas and bulk aqueous phases. Field studies trying to explain the observed HONO and $NO_x$ concentrations over the oceans postulate enhancement factors (EF) for $pNO_3^-$ photolysis rate relative to that of $HNO_3$ of 10–500 (Ye et al., 2016b, 2017a; Reed et al., 2017; Kasibhatla et al., 2018; Zhu et al., 2022; Andersen et al., 2022). Similar EFs

have also been observed in laboratory studies of photolysis of $pNO_3^-$ in ambient aerosols from urban and remote areas (Ye et al., 2017b; Bao et al., 2018; Gen et al., 2019). The high EFs could reflect the higher absorption cross-sections and quantum yields for $NO_3^-$ molecules at the surface of the particles (Zhu et al., 2008, 2010; Du and Zhu, 2011; Nissenson et al., 2010). The fraction of $NO_3^-$ at the surface is larger in the presence of halides in sea salt aerosols (Wingen et al., 2008; Richards-Henderson et al., 2013; Zhang et al., 2020). Other factors that could contribute to higher EFs include high aerosol $[H^+]$ (Scharko

et al., 2014; Mora Garcia et al., 2021) and the presence of organic species that can act as photosensitizers, H-donors, electron donors, or promote secondary reactions (Ye et al., 2019; Mora Garcia et al., 2021). Laboratory studies on $NaNO_3$ and $NH_4NO_3$ particles find EFs of less than 10 (Shi et al., 2021), suggesting that aerosol composition is an important factor in the photolysis rate of $pNO_3^-$. The relative yields of $HONO:NO_2$ in Reactions (R9a) and (R9b) also vary substantially in laboratory results. Ye et al. (2016) founds relative yields for $HONO:NO_2$ ranging from 1:1 to 30:1, with lower values for marine aerosol samples and



higher values for urban samples. Bao et al. (2018) found median relative yields for HONO:NO$_2$ of 3.5:1 for aerosol samples from Beijing.

Our baseline simulation assumes EFs of 10–100 depending on the relative amount of pNO$_3^-$ and sea salt aerosols (Eq. 1), and a HONO:NO$_2$ yield of 2:1 following Kasibhatla et al. (2018). Figure 5 shows the spatial distribution of EFs at the surface and

as a function of altitude. The simulated EF decrease from 100 in the MBL to less than 30 over the continents, where much of the pNO$_3^-$ is present as NH$_4$NO$_3$. The values over the oceans are consistent with EFs required to explain high daytime HONO concentrations (more than 10 pptv) observed over the oceans (Ye et al., 2016b; Andersen et al., 2022). Kasibhatla et al. (2018) found that an EF of 100 and a HONO:NO$_2$ yield of 15:1 were needed in GEOS-Chem to reproduce the observed diurnal cycle of HONO at Cape Verde, although EFs of 25–50 and HONO:NO$_2$ yield of 2:1 were sufficient to explain the NO$_x$ observations.

Romer et al. (2018) suggested an upper limit for the EF of 30, arguing that higher values would lead to inconsistency between the calculated steady state NO$_x$/HNO$_3$ ratios and observations from seven aircraft campaigns. Most of these campaigns were over or near continents in the northern midlatitudes, where EFs in our simulation are also generally low. In the northern midlatitudes, EFs decrease with altitude reflecting the increase in the fraction of pNO$_3^-$ present as NH$_4$NO$_3$ relative to that present on sea salt aerosols. There is little change in the EFs with altitude elsewhere.


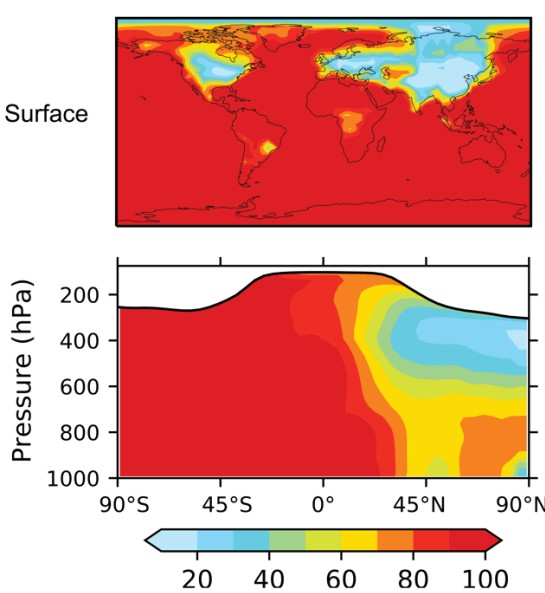

**Figure 5. Annual mean (2015) enhancement factors (EFs) for the photolysis frequency of pNO$_3^-$ with respect to the photolysis frequency of HNO$_3$ in our baseline simulation. The top panel shows EFs at the surface and the bottom panel shows the zonal mean EFs. The EF for fine pNO$_3^-$ varies from 10 to 100 according to Eq. (1) and that for coarse pNO$_3^-$ is set at 100. The values shown here**

**are the concentration weighted average EFs for total (fine + coarse) pNO$_3^-$. The zonal mean EFs are calculated as pNO$_3^-$ concentration weighted averages for the band of grid cells in each altitude and latitude bin. The white shading in the zonal mean plots denotes the stratosphere.**





pNO$_3^-$ concentrations were measured by the AMS and SAGA instruments during ATom and were found to be very low. The
median AMS and SAGA measured pNO$_3^-$ concentrations below 6 km were 10 ng sm$^{-3}$ and 44 ng sm$^{-3}$, with organic nitrates
constituting the majority of the AMS pNO$_3^-$ (Nault et al., 2021; Hodzic et al., 2020; Guo et al., 2021b). The median inorganic
pNO$_3^-$ concentrations in GEOS-Chem were 2.1 ng sm$^{-3}$ in the fine mode and 1.8 ng sm$^{-3}$ in the coarse mode. GEOS-Chem
overestimated fine mode pNO$_3^-$ concentrations in the northern midlatitudes in winter, likely because of overestimate in HNO$_3$
concentrations and aerosol pH compared to the ATom measurements (Travis et al., 2020; Luo et al., 2020; Nault et al., 2021),
but its effect on the NO$_x$ source from pNO$_3^-$ photolysis is smaller since the EF for fine mode pNO$_3^-$ photolysis decreases at
higher pNO$_3^-$ concentrations (Eq. 1). Much of the coarse mode pNO$_3^-$ measured by SAGA during ATom is associated with
dust, and probably has a lower EF than that of pNO$_3^-$ on sea salt aerosols (Andersen et al., 2022).

Including pNO$_3^-$ photolysis in the model significantly increases modeled NO$_x$ concentrations below 6 km and improves
agreement with the NO observations (Fig. 3) and with the PSS NO$_2$ concentrations inferred from NO observations (Fig. 4).
The largest increase is in the tropics (30ºS–30ºN), where pNO$_3^-$ photolysis is faster because of high actinic flux and high EFs,
and because the NO$_x$ source from PAN decomposition is small because of low concentrations at warm temperatures (Moxim
et al., 1996; Fischer et al., 2014). The effect of pNO$_3^-$ photolysis is generally smaller above 6 km because of lower pNO$_3^-$
concentrations, except in the midlatitudes in spring when pNO$_3^-$ concentrations are high and there is sufficient actinic flux.
GEOS-Chem NO$_2$ concentrations are slightly higher than the PSS NO$_2$ concentrations in the upper troposphere, because of
higher NO concentrations and higher ozone concentrations driving down the NO/NO$_2$ ratios in the model. The ozone
concentrations in the upper troposphere in the model are on average 20 ppbv higher than the ATom observations. Travis et al.
(2020) had also reported a similar overestimate in ozone concentrations in GEOS-Chem in the upper troposphere for ATom.

Figure 4 also shows the NO$_2$ profiles simulated by the GMI, TM5, and CAMS models, and Figure 6 compares the NO$_2$ column
density and AMFs for the PSS and the modeled NO$_2$ profiles. The TM5 and CAMS results are available only for August, so
the NO$_2$ column density and AMFs for August are shown separately. The campaign average (all seasons) NO$_2$ column density
is 2.4×10$^{14}$ molec cm$^{-2}$ in our baseline GEOS-Chem simulation compared to 1.9×10$^{14}$ molec cm$^{-2}$ for PSS NO$_2$, and the
corresponding AMFs are about equal (1.80). In comparison, the NO$_2$ column density in the simulation without pNO$_3^-$
photolysis is 1.5×10$^{14}$ molec cm$^{-2}$. GMI NO$_2$ concentrations are much lower than the PSS NO$_2$ concentrations below 4 km,
similar to the GEOS-Chem simulation without the pNO$_3^-$ photolysis source, and generally higher than the PSS NO$_2$
concentrations in the upper troposphere. The campaign average NO$_2$ column density in GMI is 1.4×10$^{14}$ molec cm$^{-2}$ and the
AMF is 2.2. GMI NO$_2$ concentrations are consistent with PSS NO$_2$ in the northern midlatitudes in February and in the southern
midlatitudes in August, even though GMI does not include NO$_x$ formation from pNO$_3^-$ photolysis. This is likely because GMI
does not include NO$_x$ loss through the hydrolysis of NO$_3$ and N$_2$O$_5$ in clouds (Holmes et al., 2019) or the formation of halogen
nitrates (Wang et al., 2021). The TM5 and CAMS models slightly overestimate the PSS NO$_2$ columns. Overall, the difference



in NO$_2$ column densities among the four models is ~1×10$^{14}$ molec cm$^{-2}$, indicating that the uncertainty in the NO$_2$ column retrievals associated with errors in modeled tropospheric NO$_2$ columns over clean areas is relatively small. The commonly assumed uncertainty in the tropospheric NO$_2$ columns over clean areas in the stratospheric-tropospheric separation step of satellite NO$_2$ retrievals is 2×10$^{14}$ molec cm$^{-2}$ (Bucsela et al., 2013; Boersma et al., 2018). The difference among the models in the AMFs is ~20%, which is slightly higher than the assumed uncertainty of 10% in the QA4ECV NO$_2$ column retrievals associated with the *a priori* profiles but still lower than the uncertainty associated with NO$_2$ spectral fitting and stratospheric separation in remote regions (Boersma et al., 2018).

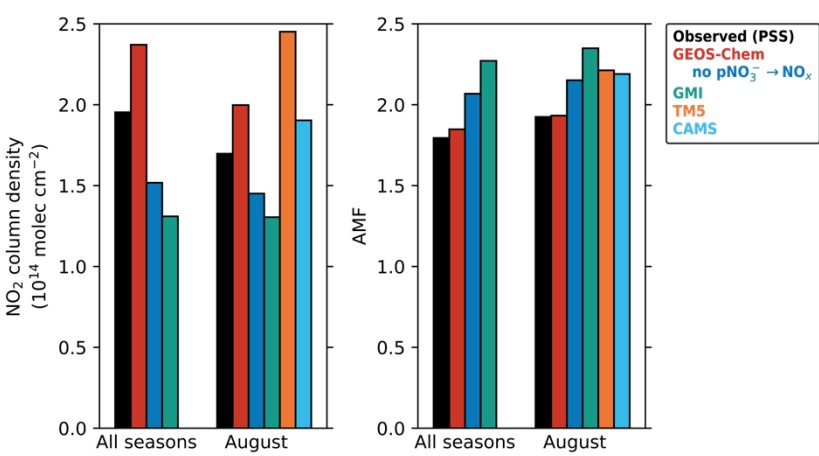

**Figure 6. Tropospheric NO$_2$ column densities and NO$_2$ air mass factors (AMFs) over the Pacific and Atlantic Oceans during ATom. The observed values are based on NO$_2$ profiles calculated using the photochemical steady state (PSS) with local measurements of NO concentrations and other quantities following Eqs. (2) and (3). Model results are from our baseline GEOS-Chem simulation, GEOS-Chem simulation without pNO$_3^-$ photolysis, GMI, TM5, and CAMS, sampled along the flight tracks. Values shown are the average for all four ATom deployments and the average for the August deployment, as the TM5 and CAMS NO$_2$ profiles are available only for August. AMFs are calculated using the NASA OMI NO$_2$ v4.0 scattering weights following Eq. (5), with the geometric AMF (AMF$_G$) values of 2.6 for 30ºS–30ºN and 3.7 for 30ºS–60ºS and 30ºN–60ºN.**

### 3.3 Effect of pNO$_3^-$ photolysis on global NO$_x$, OH, and ozone concentrations

Figure 7 shows the change in the annual mean NO$_x$, OH, and ozone concentrations at the surface and zonally between our baseline simulation and the sensitivity simulation without pNO$_3^-$ photolysis. pNO$_3^-$ photolysis increases NO$_x$, OH and ozone tropospheric masses in the model by 9%, 19%, and 10%, respectively. In comparison, Kasibhatla et al. (2018) found increases in the NO$_x$, OH, and ozone tropospheric masses of 1–3% in simulations that included photolysis of only coarse mode pNO$_3^-$ at an EF of 100 and increases of 3–6% when fine mode pNO$_3^-$ photolysis was also included at an EF of 25. NO$_x$ concentrations increase by a factor of 2 on average in the MBL, consistent with our results for the ATom campaign. There is little increase in



the northern extratropical MBL, as PAN concentrations are high and provide the main source of $NO_x$ in the region. There is little change in surface $NO_x$ concentrations over continents as local emissions dominate the $NO_x$ source. $NO_x$ concentrations decrease slightly over some regions because the increase in OH concentrations shortens the $NO_x$ lifetime. The increase in $NO_x$ concentrations in the tropics and subtropics is limited mostly to the MBL, since $pNO_3^-$ concentrations are low at higher

altitudes. In the free troposphere of the northern midlatitudes, $pNO_3^-$ photolysis increases $NO_x$ concentrations by just 20%, because $pNO_3^-$ concentrations and EFs for $pNO_3^-$ photolysis are generally low (Fig. 5). The effect of $pNO_3^-$ photolysis is larger in spring, when there is a seasonal peak in $pNO_3^-$ concentrations in the model. There is a large increase in $NO_x$ concentrations over Antarctica, as there are few other $NO_x$ sources in the region in the model. Our simulations do not include snow $NO_3^-$ photolysis, which is an important source of $NO_x$ in the region (Zatko et al., 2016).


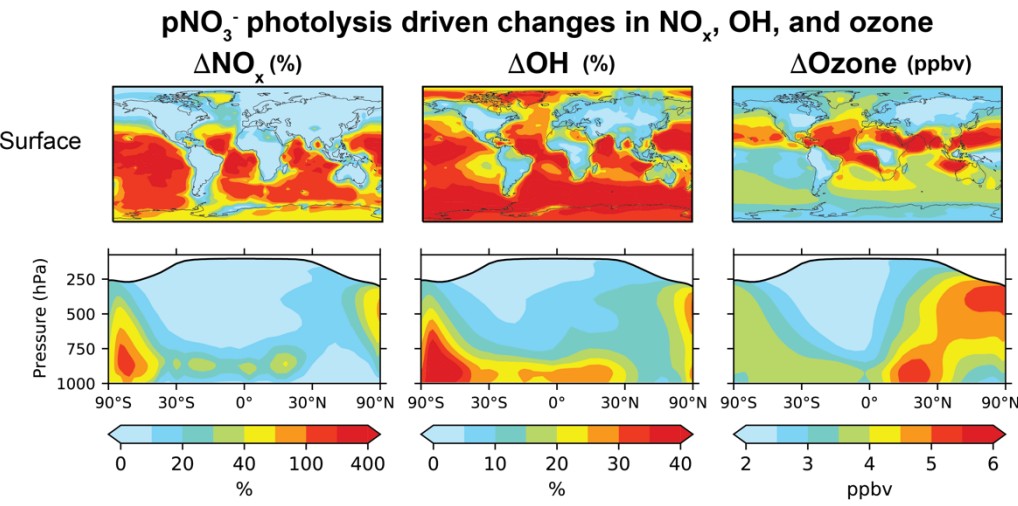

**Figure 7. Annual mean (2015) changes in $NO_x$, OH, and ozone concentrations from including $pNO_3^-$ photolysis in GEOS-Chem. The top panels show changes at the surface and the bottom panels show the zonal means. Changes in $NO_x$ and OH are shown as percent change of concentrations in our baseline simulation relative to that in the sensitivity simulation without $pNO_3^-$ photolysis, and**
**changes in ozone are shown as differences in concentrations (in ppbv) between the two simulations. The white shading in the zonal mean plots denotes the stratosphere.**

$pNO_3^-$ photolysis increases the production of OH and ozone because of the increase in $NO_x$ concentrations in low-$NO_x$ regions, where OH and ozone production are more sensitive to $NO_x$ concentrations. OH is also produced by the photolysis of HONO
released to the gas phase during $pNO_3^-$ photolysis (Reaction R10), which would be important source of OH in winter when OH production from Reaction (R6) is slow (Elshorbany et al., 2012). The increase in OH concentrations is particularly large (~30%) in the MBL. Travis et al. (2020) showed that the GEOS-Chem OH concentrations from a simulation without $pNO_3^-$ photolysis are consistent with the ATom observations, but they also found an underestimate in the modeled OH reactivity in the lower troposphere due to missing VOCs in the model. The source of these VOCs is likely oceanic (Thames et al., 2020)
and would depress model OH, which could then be compensated by an increase in the OH source from $pNO_3^-$ photolysis. The



OH increase implied by pNO$_3^-$ recycling decreases the global atmospheric methane lifetime from 8.0 years to 7.0 years, worsening the agreement with the value of $9.1 \pm 0.9$ years inferred from the methylchloroform proxy (Prather et al., 2012), but again this could be compensated by a model underestimate of OH reactivity (Travis et al., 2020; Kim et al., 2022).

pNO$_3^-$ photolysis increases surface ozone concentrations by 3.6 ppbv on average at the surface, and up to 8 ppbv in the tropics and subtropics. In the northern extratropics, the ozone increase is small at the surface, but about 5 ppbv in the free troposphere, reflecting the spatial pattern of increase in NO$_x$ concentrations. Wang et al. (2021) recently evaluated the GEOS-Chem ozone simulation with ozonesonde observations and found an underestimate in simulated free tropospheric ozone of 5–15 ppbv in the northern hemisphere and up to 5 ppbv in the southern hemisphere, depending on whether halogen chemistry was included

or not. Including the NO$_x$ source from pNO$_3^-$ photolysis improves GEOS-Chem's ozone simulation. We will examine this further in a future publication.

### 3.4 Primary sources of NO$_x$ in the free troposphere

NO$_x$ in the free troposphere originates from a variety of primary sources with differing spatial and seasonal characteristics. The sources include in situ emissions from lightning and aircraft, uplifting of NO$_x$ emitted from surface sources, and

downwelling of stratospheric NO$_y$ produced from the photolysis of N$_2$O. Lightning is the main in situ source of NO$_x$ in the free troposphere globally (Table 2), but it is concentrated over continents and has a strong seasonality in the midlatitudes. Aircraft emissions are largest in the northern midlatitudes, and while most of the aircraft emissions are over land, there are significant emissions over the northern Atlantic and Pacific oceans (Simone et al., 2013). Surface emissions are widely distributed over the tropics and the northern midlatitudes, but their transport to the free troposphere would vary seasonally.

Here we use GEOS-Chem to determine the relative importance of these primary sources for NO$_x$ in the free troposphere.

Figure 8 shows the vertical profiles of NO$_x$ over the Pacific and Atlantic Oceans and the contiguous US for February and August separately for surface emissions (fuel combustion, fires, and soils and fertilizer use), aircraft emissions, and lightning emissions. We focus on the tropospheric sources and exclude the stratospheric NO$_y$ source from N$_2$O because it is small in the

global troposphere, although it could be important in the upper troposphere in the high latitudes in summer (Levy et al., 1999). The source contributions are derived from three sensitivity simulations with small (20%) perturbation to each source in turn and are calculated as $([NO_x]_0 - [NO_x]_i)/\sum_{i=1}^{3}([NO_x]_0 - [NO_x]_i)$, where $[NO_x]_0$ is the NO$_x$ concentrations in the baseline simulation and $[NO_x]_i$ is the NO$_x$ concentration in the sensitivity simulation $i$. The source contributions for the northern midlatitudes are shown separately for February and August, but for the tropics and southern midlatitudes the February and

August average is shown.

Over the northern midlatitude oceans, in February, most of the NO$_x$ in the free troposphere is supplied by surface and aircraft sources. Both sources contribute equally (42%) to the free tropospheric NO$_x$ column, but surface emissions are dominant below



6 km and aircraft emissions above 6 km. In August, lightning is the dominant source of NO$_x$, supplying 55% of the NO$_x$ column

in the free troposphere. Aircraft emissions contribute 33% but are the major source of NO$_x$ between 10 and 12 km. Aircraft

emissions account for the higher NO$_x$ concentrations in the upper troposphere over the northern midlatitudes than over the

tropics and southern midlatitudes. Lightning is the dominant source of NO$_x$ in the tropics and the southern midlatitudes,

supplying 62–68% of the free tropospheric NO$_x$ column, with surface sources supplying 18–30% of NO$_x$. However, the

contribution of surface sources may be underestimated in the model. Bourgeois et al. (2021) showed that the highest NO$_y$ and

ozone concentrations during ATom were observed in polluted air from biomass burning sources, but this was not reproduced

by GEOS-Chem and other models, reflecting model uncertainties in biomass burning emission inventories, plume injection

heights, and export efficiency of biomass burning emissions to the free troposphere.

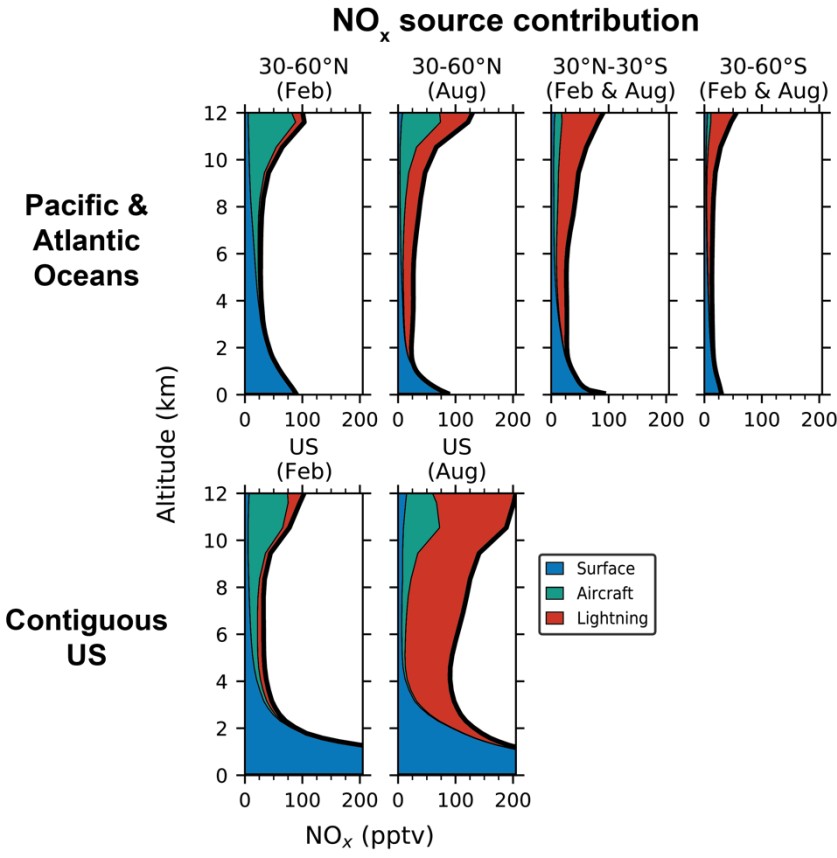

**Figure 8. Vertical profiles of NO$_x$ concentrations from three primary source categories over the Pacific and Atlantic oceans and over the contiguous US in GEOS-Chem. The source categories include surface emissions (fuel combustion, soil, fertilizer use, and fires), aircraft emissions, and lightning emissions. The stratospheric NO$_y$ source from N$_2$O is not included here. The profiles for the Pacific and Atlantic Oceans are means for the regions sampled in ATom (160ºE–160ºW and 25º–50ºW in the northern hemisphere and 160ºE–160ºW and 0º–30ºW in the southern hemisphere) and are separated into three latitude bands. The northern midlatitude (30º–**

**60ºN) profiles are further separated by month (February and August), while the other profiles are the average for the two months. The profiles for the contiguous US (defined as 25º–50ºN and 65º–130ºW) are shown separately for February and August.**



We compare the $NO_x$ source contribution over the northern midlatitude oceans to that over the contiguous US. We find that the $NO_x$ sources over the oceans and the US are similar in winter. Surface and aircraft sources each supply about 40% of $NO_x$ in the free troposphere in February over the US, with surface sources dominating below 4 km and aircraft sources in the upper troposphere. In August, lightning emissions supply 73% of the $NO_x$ in the free troposphere over the US, much more than in winter and over the oceans. Previous modeling studies have also found lightning to be the main source of $NO_x$ in the tropics and southern midlatitudes, and a seasonal change in the main source in the northern midlatitudes from lightning in summer to surface and aircraft emissions in winter (Lamarque et al., 1996; Levy et al., 1999). But the contribution of aircraft emissions to free tropospheric $NO_x$ in our simulation is higher than in these previous studies, reflecting a nearly two-fold increase in global aircraft $NO_x$ emissions in the past three decades (Hoesly et al., 2018).

### 3.5 Implications for the retrieval and interpretation of satellite $NO_2$ data

We showed that the previously reported model underestimate of $NO_2$ concentrations in the upper troposphere over the US can be attributed to interference in the $NO_2$ measurements, and that when compared with the measured NO and PSS $NO_2$ profiles, the modeled $NO_2$ profiles in the free troposphere are consistent with the $SEAC^4RS$, DC3, and ATom observations. This increases our confidence in the modeled $NO_2$ profiles and here we use them to examine the importance of the free troposphere in the retrieval and interpretation of satellite $NO_2$ data over the US. Figure 9 shows the GEOS-Chem vertical profiles of the $NO_2$ number density in the early afternoon (OMI and TROPOMI overpass time) over the contiguous US for summer and winter of 2015. The results are from our baseline simulation, but there is little difference in the $NO_2$ profiles between our baseline simulation and the simulation without $pNO_3^-$ photolysis over the US (Fig. 1), except in spring when the $NO_2$ concentrations in the free troposphere are about 10% higher due to the $pNO_3^-$ photolysis source.

In summer, simulated $NO_2$ partial columns in the boundary layer and the free troposphere are $6.9 \times 10^{14}$ molec $cm^{-2}$ and $5.8 \times 10^{14}$ molec $cm^{-2}$, respectively. In comparison, the simulated wintertime $NO_2$ partial columns are $15.4 \times 10^{14}$ molec $cm^{-2}$ and $1.9 \times 10^{14}$ molec $cm^{-2}$ in the boundary layer and the free troposphere. The boundary layer $NO_2$ column is higher in winter because of longer $NO_x$ chemical lifetimes (Kenagy et al., 2018; Shah et al., 2020) and slower ventilation to the free troposphere, while the free troposphere $NO_2$ column is higher in summer because of lightning emissions (Fig. 7). The GEOS-Chem summertime $NO_2$ column density in the free troposphere over the US is about three times higher than the PSS-inferred $NO_2$ column over the oceans during ATom. But, in winter, the free tropospheric $NO_2$ column over the US is similar to that over the oceans, indicating little contrast in background $NO_2$ between the US and surrounding oceans. There is also little contrast in the free tropospheric $NO_2$ sources in winter between the continents and the surrounding oceans (Fig. 7). This reflects the longer lifetime of $NO_x$ in winter because of slower photochemistry and increased recycling of $NO_3^-$ to $NO_x$ over the oceans through $pNO_3^-$ photolysis. It also implies that ATom observations over the northern midlatitudes in February could be used to estimate



the free tropospheric NO₂ concentrations over the US in winter, in the absence of aircraft observations over land that probe the full height of the winter troposphere. Marais et al. (2018) had compared the GEOS-Chem NO₂ concentrations at 6–10 km with those derived from the OMI cloud-sliced product (Choi et al., 2014) for 2005–07 and found that GEOS-Chem underestimates NO₂ concentrations over North America in winter by about a factor of 3. The successful simulation of the measured NO and the PSS NO₂ concentrations over the northern midlatitudes in winter during ATom suggests that there is little bias in the free

tropospheric NO₂ concentrations in the model, and that the underestimate with respect to the OMI observations likely reflects uncertainties in the cloud-slicing technique. The simulation used by Marais et al. (2018) did not include the NOₓ source from pNO₃⁻ photolysis, but its effect on NO₂ concentrations over continents is not large (Fig. 6). The global aircraft NOₓ emissions in their simulation were 30% lower than those in our simulation, reflecting conditions for the year 2006.

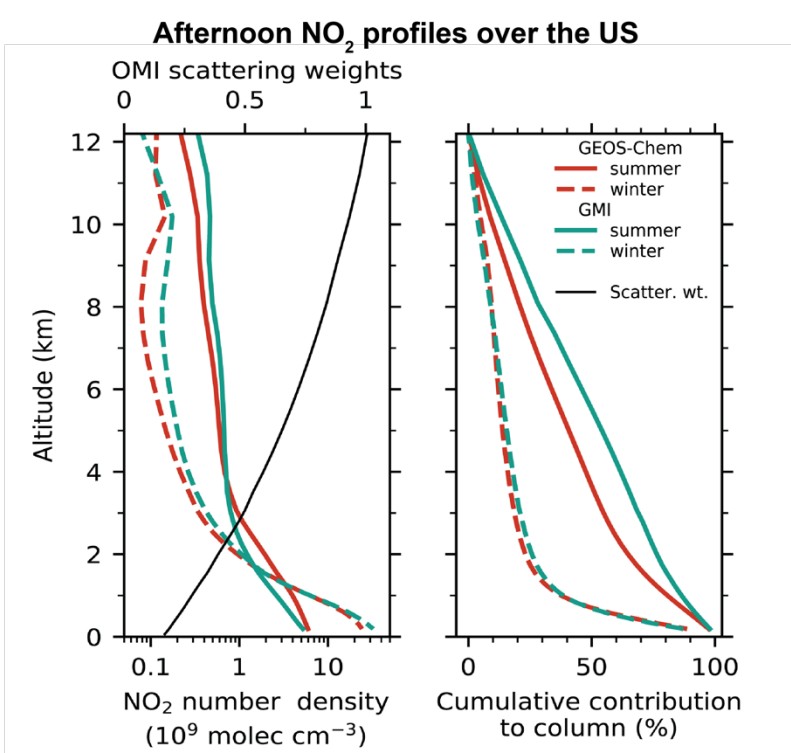


**Figure 9. Seasonal mean vertical profiles of NO₂ number density over the contiguous US and cumulative percent contributions to tropospheric NO₂ columns as would be observed by the OMI satellite instrument. The left panel shows the afternoon NO₂ profiles simulated by GEOS-Chem and GMI for summer (June, July, and August) and winter (December, January, and February) for the**
**year 2015. The NO₂ number density is shown on a log scale. Also shown in the left panel is the mean profile of the scattering weights from the NASA OMI NO₂ (v4.0) retrievals averaged over summer and winter for clear scenes (cloud fraction < 0.1) and dark surfaces (surface albedo < 0.3). There is little difference in the scattering weight profile between winter and summer. The right panel shows the cumulative percent contribution from NO₂ at different altitudes to the tropospheric NO₂ columns as measured by OMI for summer and winter. It is calculated using Eq. (6).**





Figure 9 also shows the afternoon NO$_2$ number density from the GMI model. The GEOS-Chem and GMI profiles have consistent shapes, but there are some differences in the free tropospheric NO$_2$ concentrations, likely because of differences in NO$_x$ oxidation chemistry, as described in Section 3.3, and differences in the lightning and aircraft NO$_x$ emissions. While both models use similar parameterizations for lightning emissions (Allen et al., 2010; Murray et al., 2012), GMI assumes a higher NO yield over the US (500 moles per flash north of 26ºN and 250 moles per flash south of it) than GEOS-Chem (500 moles per flash north of 35ºN and 260 moles per flash south of it). Aircraft emissions in GMI are from an older inventory (Duncan et al., 2007) with global NO$_x$ emissions of 0.56 TgN a$^{-1}$ compared to 1.2 TgN a$^{-1}$ in our simulation.

We calculate the seasonal AMFs corresponding to the GEOS-Chem and GMI NO$_2$ profiles (Eq. 5) to determine the effect of different a priori profiles on the retrieved NO$_2$ columns. As before, we use scattering weights from the NASA OMI NO$_2$ retrieval (v4.0) for clear sky and dark surfaces. The scattering weight profile over the US is also shown in Fig. 9. There is little seasonal difference in the scattering weight profile because we exclude cloudy scenes and bright surfaces, but AMF$_G$ is higher in winter (3.7) than in summer (2.6) because of higher solar zenith angles. In summer, the AMF calculated using the GEOS-Chem profile is 1.14, compared to 1.33 calculated with the GMI profile. In winter, the AMFs from the two models are nearly identical (1.0). The AMFs are lower in winter than in summer because of higher NO$_2$ concentrations in winter in the boundary layer where satellite measurements are less sensitive. GEOS-Chem NO concentrations in the free troposphere were about 2 times higher than the measurements during SEAC$^4$RS and DC3 (Fig. 1). If we decrease the GEOS-Chem NO$_2$ number density in the free troposphere by half in summer, then the AMF decreases to 0.98. Boersma et al. (2018) estimated a single-pixel uncertainty in the QA4ECV retrieval AMFs over the US of 20% in summer and 25% in winter. The largest contributions to the AMF uncertainty were associated with surface albedo and cloud properties; the uncertainty associated with the NO$_2$ profiles was assumed to be 10% in all seasons. We find that the uncertainty in the NO$_2$ profiles is higher than 10% in summer, because of uncertainty in the free tropospheric NO$_2$ columns.

The right panel of Fig. 9 shows the cumulative contribution from different altitudes to the tropospheric NO$_2$ columns as would be measured by OMI. It is calculated as:

$$\Gamma(z) = 100 \times \frac{\int_z^{z_t} w(z)\,n(z)\,dz}{\int_0^{z_t} w(z)\,n(z)\,dz}, \qquad (6)$$

where $\Gamma(z)$ is the percent cumulative contribution to the tropospheric NO$_2$ column at altitude $z$, $z_t$ is the tropopause altitude, and $w(z)$ and $n(z)$ are the vertical profiles of the scattering weight and the NO$_2$ number density. The contribution of the free troposphere to NO$_2$ columns is significantly higher in summer than in winter. In summer, the free troposphere contributes 65% of the tropospheric NO$_2$ column over the US in GEOS-Chem (75% in GMI), whereas in winter, 75% of the NO$_2$ column resides below 2 km. The free tropospheric contribution decreases to 55% if we halve the GEOS-Chem NO$_2$ column in the free troposphere in summer. Travis et al. (2016) had also calculated a free troposphere contribution of 70–75% from the GEOS-Chem NO$_2$ profiles in SEAC$^4$RS.





The large contribution of the free troposphere to NO₂ columns affects the interpretation of satellite data in terms of NOₓ emissions. It greatly diminishes the sensitivity of the summertime NO₂ columns to changes in surface NOₓ emissions when averaged over the entire US. The free tropospheric contribution would be small over major cities, where summertime NO₂ column densities exceed $3 \times 10^{15}$ molec cm⁻² (Lamsal et al., 2021), but still needs to be accounted for. Urban NOₓ emissions and their trends are commonly derived by fitting an exponential decay function to satellite NO₂ columns downwind of the

source (e.g., Beirle et al., 2011; Lorente et al., 2019; Goldberg et al., 2021). The fitting function includes a background offset term and thus implicitly accounts for the free tropospheric background. The free tropospheric background is also accounted for when models that include lightning and aircraft NOₓ emissions are used to relate NO₂ columns to NOₓ emissions, but there is substantial uncertainty in the magnitude and distribution of lightning NOₓ emissions (Schumann and Huntrieser, 2007; Murray, 2016), which is the main source of NO₂ in the free troposphere in summer. Missing organic NOₓ chemistry in summer

would also contribute to model errors in the free tropospheric NO₂, as suggested by our SEAC⁴RS and DC3 analysis. Wintertime NO₂ columns will respond more strongly to changes in NOₓ emissions, but the uncertainty in the NO₂ retrievals associated with surface albedo and clouds is larger in winter (Boersma et al., 2018). Free tropospheric NO₂ columns are low in winter, but not negligible, and their simulation would be affected by uncertainties in aircraft emissions (Simone et al., 2013) and NOₓ chemistry involving heterogeneous reactions (McDuffie et al., 2018; Holmes et al., 2019) and halogens (Wang et al.,

2021). Better observational constraints on free tropospheric NO₂ concentrations are needed to help reduce these uncertainties.

## 4. Conclusions

    We used aircraft measurements from the SEAC⁴RS, DC3, and ATom campaigns to evaluate the vertical distribution of NOₓ in the free troposphere in the GEOS-Chem, GMI, TM5, and CAMS atmospheric chemistry models because of its importance for the simulation of tropospheric ozone and OH and for the retrieval and interpretation of satellite NO₂ column measurements.

We first examined the accuracy of the in situ NO₂ measurements in the upper troposphere using observations made in a thunderstorm outflow during the DC3 campaign. We found that the laser induced fluorescence (LIF) and the photolysis-chemiluminescence (P-CL) NO₂ measurements were significantly higher than the NO₂ concentrations calculated using the NO measurements and the NO-NO₂ photochemical steady state (PSS), and that the ozone production expected based on these NO₂ measurements was much higher than the observed ozone production. This indicates an interference in the NO₂ measurements,

presumably from HNO₄ and methyl peroxy nitrate (MPN), and can explain the underestimate in modeled NO₂ concentrations relative to measurements in the upper troposphere reported previously (Travis et al., 2016; Silvern et al., 2018).

    GEOS-Chem reproduces the shapes of the vertical profiles of the NO observations and the PSS NO₂ concentrations inferred from the NO measurements during SEAC⁴RS and DC3. The NO₂ air mass factors (AMFs) calculated using the PSS and the

GEOS-Chem NO₂ vertical profiles from SEAC⁴RS and DC3 and scattering weights from the NASA OMI NO₂ v4.0 retrievals



differed by less than 10%. However, GEOS-Chem overestimates $NO_2$ concentrations in the free troposphere over the southeastern US by about a factor of 2, and underestimates concentrations of MPN and alkyl nitrates, suggesting missing organic $NO_x$ chemistry in the model, which needs to be examined in the future.

The NO concentrations measured over the Pacific and Atlantic Oceans were reproduced by GEOS-Chem when $pNO_3^-$ photolysis was included in the model with photolysis frequencies 10–100 times higher than that of gas-phase $HNO_3$, as suggested by laboratory studies of $pNO_3^-$ photolysis and field studies of HONO sources in the marine atmosphere (Ye et al., 2016a, 2017b; Andersen et al., 2022). The average $NO_2$ column density for the ATom campaign was $1.9 \times 10^{14}$ molec $cm^{-2}$ for the PSS $NO_2$ concentrations, and $2.4 \times 10^{14}$ molec $cm^{-2}$ for GEOS-Chem with $pNO_3^-$ photolysis and $1.5 \times 10^{14}$ molec $cm^{-2}$

without. The $NO_2$ column density for the GMI, TM5, and CAMS models was between 1.4 and $2.5 \times 10^{14}$ molec $cm^{-2}$ and the $NO_2$ AMFs calculated using the PSS $NO_2$ profiles and the simulated $NO_2$ profiles differed by less than 20%. Model errors in the tropospheric $NO_2$ profiles over the remote oceans are not a major source of uncertainty in the satellite $NO_2$ retrievals. We calculated the contribution of surface, aircraft, and lightning emissions to $NO_x$ columns over the Pacific and Atlantic Oceans and over the US in GEOS-Chem, and found that lightning is the main $NO_x$ source over the tropics and southern midlatitudes,

and over the US in the summer, contributing 62–73% of the $NO_x$ columns in the free troposphere. However, aircraft emissions are the main source of free tropospheric $NO_x$ in the northern mid-latitudes in winter and in summer over the oceans.

$pNO_3^-$ photolysis increases the global tropospheric mass of $NO_x$, OH and ozone in GEOS-Chem by 9%, 19%, and 10%, respectively. $NO_x$ concentrations increase most in the tropical MBL where $NO_x$ sources from PAN are small. There is a small

increase in $NO_x$ concentrations in the free troposphere over the continents, but the increase is larger in spring, when there is a seasonal peak in $pNO_3^-$ concentrations in the model. The increase in OH concentrations would degrade the model performance relative to OH measurements in ATom, but the ATom observations also indicate an underestimate in the modeled OH reactivity in the lower troposphere (Travis et al., 2020). Ozone concentrations increased up to 8 ppbv at the surface in the tropics and subtropics, and by 5 ppbv in the free troposphere over the northern extratropics, which would largely correct the low model

bias relative to ozonesonde observations (Wang et al., 2021).

The seasonal GEOS-Chem and GMI afternoon $NO_2$ profiles over the contiguous US are largely consistent with each other and show higher boundary layer $NO_2$ columns in winter than in summer because of longer $NO_x$ chemical lifetimes and slower ventilation to the free troposphere, but higher free tropospheric $NO_2$ columns in summer because of lightning emissions. In

winter, the free troposphere contributes 25% of the $NO_2$ columns that would be observed by satellite instruments over the contiguous US, but in summer this increases to 65–75%, and weakens the sensitivity of the summertime $NO_2$ columns to changes in surface $NO_x$ emissions. Better $NO_2$ observations are needed to reduce model uncertainties in lightning $NO_x$ emissions and $NO_x$ chemistry in the free troposphere.





*Data availability:* The SEAC⁴RS aircraft measurements are available at https://doi.org/10.5067/Aircraft/SEAC4RS/Aerosol-TraceGas-Cloud (last access: 1 July 2021, SEAC⁴RS Science Team, 2014), DC3 at https://doi.org/10.5067/Aircraft/DC3/DC8/Aerosol-TraceGas (last access: 1 July 2021, DC3 Science Team, 2013), and ATom at https://doi.org/10.3334/ORNLDAAC/1925 (last access: 1 July 2022, Wofsy et al., 2021). The GMI model results for ATom are available at https://doi.org/10.3334/ORNLDAAC/1897 (last access: 1 July 2021, Strode et al., 2021). All other model
results are available on request from the corresponding author.

*Author contributions:* VS and DJJ designed the study and led the analysis. RD helped with interpreting the GEOS-Chem results. LNL, SAS, and SDS provided the GMI simulation results and KFB provided the TM5 and CAMS results. SDE and TMF provided the updated AEIC inventory. CT, JP, IB, IP, BAN, RCC, PCJ, and JLJ made the NO, $NO_2$, $NO_y$, ozone, and
$pNO_3^-$ measurements during the SEAC⁴RS, DC3, and ATom campaigns. STA, LJC, TS, and MJE helped with the $pNO_3^-$ photolysis simulation. VS and DJJ wrote the paper with input from all authors.

*Competing interests:* The authors declare that they have no conflict of interest.

*Acknowledgements:* We are grateful to the instrument teams of the SEAC⁴RS, DC3 and ATom campaigns for making their
data freely available. We thank Tom Ryerson (NOAA) for contributing to the NO, $NO_2$, $NO_y$, and $O_3$ measurements in the three campaigns, and Eloïse Marais (U. College London) and Sunny Choi (NASA GSFC) for helpful discussions. This product/document has been created with or contains elements of Base of Aircraft Data (BADA) Family Release which has been made available by EUROCONTROL to MIT. EUROCONTROL has all relevant rights to BADA. ©2019 The European Organisation for the Safety of Air Navigation (EUROCONTROL). All rights reserved. EUROCONTROL shall not be liable
for any direct, indirect, incidental, or consequential damages arising out of or in connection with this product or document, including with respect to the use of BADA. GMI is supported by the NASA Modeling, Analysis, and Prediction (MAP) program.  GMI simulations used computational resources from the NASA High-End Computing (HEC) Program through the NASA Center for Climate Simulation (NCCS). JLJ and PCJ were supported by NASA Grant 80NSSC21K1451.

*Financial support:*  This work was supported by the NASA Aura Science Team and by the US EPA Science To Achieve Results (STAR) program.

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
