# Peer review of "Nitrogen oxides in the free troposphere: Implications for tropospheric oxidants and the interpretation of satellite NO2 measurements"

_EGUsphere, 2022_

## Referee Comment (RC3)

The goal of this work is to improve our understanding of $NO_x$ in the free troposphere, and implications for atmospheric oxidation and interpretation of satellite $NO_2$. Overall, this paper looks at how well four atmospheric chemistry models simulate background $NO_x$ leveraging observations from three aircraft campaigns. The authors show that observed ozone is consistent with steady state calculations using observed NO, and it is likely that upper tropospheric $NO_2$ measurements have interferences from thermal decomposition of $NO_x$ reservoirs at cold temperatures. The authors find that lightning and aircraft emissions are the main contributors to the free tropospheric contribution of $NO_2$ to the total column but differ in their seasonality. The model underestimates NO (and the $NO_2$ column) during the ATom campaign over the remote oceans, which the authors suggest is due to missing photolysis of particulate nitrate.

This paper could be a useful contribution to the field and to those trying to use $NO_2$ satellite information combined with models to infer emissions. I recommend the authors make the following minor revisions before publication in ACP.

**Major Comments**

One concern is that this paper is relied on for evidence of particulate nitrate photolysis "Andersen, S. T., Carpenter, L. J., Reed, C., Lee, J. D., Chance, R., Sherwen, T., Vaughan, A. R., Bloss, W. J., Sommariva, R., Nott, G., Neves, L., Read, K., Heard, D. E., Seakins, P. W., Whalley, L. K., Boustead, Fleming, L. T., Stone, D., and Fomba, K. W.: Extensive field evidence for the release of HONO from the photolysis of nitrate aerosols, Sci. Adv. (in review), 2022." but it is not yet available. As photolysis of particulate nitrate is currently poorly constrained, it appears that this paper with help support the inclusion of this pathway in models.

The authors need to better defend the solution of particulate nitrate photolysis, and not underestimated transport of NOx in alkyl or peroxy nitrates, or direct NO emissions from the ocean, particularly since the authors discuss that underestimated MPN and alkyl nitrates in the model suggests missing chemistry.

The authors should also be clearer about their recommendations for how $NO_2$ observations should be treated in the future. Should they be ignored in favor of PSS, or should measurements focus on the improvements made during Bradshaw et al., 1999?

Finally, the authors should discuss recommendations for observing emissions trends given the importance of the free tropospheric contribution to columns, for example by only looking at regions with column amounts over a certain threshold. They haven't quantified for us how important the free tropospheric background is to the column over cities and whether an approach removing columns with a significant background contribution could help to clarify trends in surface NOx emissions.

**Minor comments**

Line 129 – This statement is confusing "Silvern et al. (2018) showed that using the observed $NO_2$ vertical profile from SEAC$^4$RS in the NASA $NO_2$ column retrieval for the OMI satellite instrument decreases the retrieved NO2 columns over the southeastern US by 30%, suggesting

the possibility of a systematic bias in the NO2 column retrievals." You have just told us that $NO_2$ measurements have interferences, so likely the observed profile is biased, not the retrieval. Please clarify.

In Bradshaw et al., 1999, observations matched photostationary steady state, using a "a highly modified photofragmentation two-photon laser-induced fluorescence (PF-TP-LIF) instrument." Could you explain how these modifications differ from current LIF techniques and provide some guidance on whether the modifications from Bradshaw et al., 1999 should be employed in the future to better measure $NO_2$?
    Similarly, can you explain why the MPN corrections in the SEAC4RS LIF observations were insufficient?

Line 248 – How do you justify that sea salt is internally mixed with sulfate/ammonium/nitrate in the fine mode?

Line 390 – Please clarify. Is this 'unheated' channel the one you are saying still has a measurement bias, presumably because the channel is still warmer than ambient temperatures due to being on the inside of the aircraft?

Line 410 – I am curious if your simulations with photolysis of particulate nitrate produced appreciable during SEAC4RS/DC3 as SENEX observations showed HONO outside of direct sources was negligible (< 15 ppt)
https://agupubs.onlinelibrary.wiley.com/doi/abs/10.1002/2016JD025197.

Line 430 – How does this finding change the conclusion of Travis et al., 2016 that the model $NO_2$ columns were inconsistent with other constraints on emissions?

Line 435 – Does it really make sense to discuss a 'global background of $NO_2$' given its short lifetime of only hours? Also, "background" is sometimes used to mean non-anthropogenic. Possibly consider different terminology.

Fisher et al., 2018 (https://agupubs.onlinelibrary.wiley.com/doi/10.1029/2018JD029046) suggest that the underestimated NO could be direct emissions from seawater. Discuss whether this is a possible alternative to photolysis of particulate nitrate.

Line 474 – As you are also simulating $HNO_3$ and PAN, it would be more consistent to compare your own simulations to $HNO_3$ and PAN, and then you could avoid a citation to a presentation (Zhai et al., 2022) rather than peer-reviewed literature. The authors should also show comparisons with observed particulate nitrate and show whether photolysis improves or degrades that simulation.

Line 474 – How do you know there is no overestimate of $NO_x$ sinks? You haven't shown us this. Or do you just mean $NO_2$ + OH is not overestimated which is why you cite Travis et al., 2022 for OH?

Line 476 – Also Henderson et al., 2012 - https://acp.copernicus.org/articles/12/653/2012/ and Seltzer et al., 2015 https://acp.copernicus.org/articles/15/5973/2015/acp-15-5973-2015.pdf.

Line 585 – Can you discuss again the biases in your GEOS-Chem simulation of PAN in the context of this discussion? Is PAN underestimated in the MBL?

Line 620 – This conclusion is without addressing model underestimates of OH reactivity. Please discuss how you would expect model ozone to change if OH reactivity was increased in the model.

Line 694 – Why would estimating free tropospheric concentrations in February be beneficial? What would be the application for such an estimate?

Line 718 – What are the differences in the reaction rates for $NO + O_3$ and $NO_2 + OH + M$ in GMI vs. GEOS-Chem?

Figure 1 – Is there a reason to put NO on the same scale as $NO_2$? It would be easier to see the discussed NO biases if the scale was reduced.

Line 732 – Is there a reason this decrease would impact only the free troposphere and not the boundary layer?

Line 765 – I assume then that the higher spatial resolution of TEMPO will allow for a better examination of trends in emissions moving forward as it will be less sensitive to FT background. Also, why haven't studies only focused over urban regions to look at trends?

Line 776 – Can you clarify exactly what needs to be done for future $NO_2$ measurements? Reconsider the mount of MPN that might interfere? Make the modifications described in Bradshaw et al., 1999?

Line 784 – This again makes me curious about peroxy nitrates (PAN etc) as well, and whether the model underestimates $NO_x$ transport that could be part of the underestimate in ATom NO. You say on line 299 that PAN is small, but also the perturbation in NO needed is small.

Line 805 – But again, how would you expect the ozone response to change if OH reactivity was increased?

Line 813 – Is there any reason to suggest focusing analysis of NOx trends/emissions on regions where the $NO_x$ column is high enough to have low uncertainty from the large free tropospheric background?

---

## Author Response (AR1)

**Response to comments by Referee # 1**

This manuscript contains important analyses with regard to observations and model calculations of NOx in the free troposphere. The paper convincingly proves that NO2 from both laser-induced fluorescence and photolysis/chemiluminescence instruments have significant high biases due to interferences from other NOy species. NO2 calculated from photostationary state (PSS) assumptions likely yield a better estimate. However, the GEOS-Chem model overestimates the PSS-NO2 in the fre e troposphere in the southeast US during the SEAC4RS and DC3 experiments. In the remote free troposphere, the model underestimates NO during ATom, but inclusion of photolysis of particulate nitrate greatly improves the simulations. The implications of these findings for NO2 satellite retrievals are discussed. As found in previous studies, lightning is noted as the primary NOx source to the free troposphere over the tropics and southern midlatitudes in all seasons and over the US in summer. The free tropospheric component of the NO2 column over the US in summer (65%) is sufficiently large to make surface emissions estimates in this season difficult. This is an important conclusion of the manuscript. The paper is very well written and should be published with just a few minor revisions as noted below:

We thank the reviewer for their thoughtful and supportive comments. Our response to their specific comments is as follows:

**line 131: ....column retrievals, if the airborne measurements are assumed to be correct.**
This sentence contradicted our earlier point about issues with the $NO_2$ measurements and we have deleted it.

**line 167: NOy/NO > 3 seems like this would be aged emissions, not fresh. Maybe this should be < 3 ?**
Thank you for catching this typo. It is indeed "NOy/NO <3." We've corrected this now.

**line 563: "....errors in modeled tropospheric NO2 columns over clean areas in relatively small." This doesn't seem correct based on the model results shown in Figure 6. The difference between models is ~1 x 10^14 and the PSS-based NO2 column is ~1.9 x 10^14. Wouldn't this imply an uncertainty greater than 50%?**
This was indeed a bit confusing. We were trying to point out that the model error in $NO_2$ columns over clean areas is small compared to error associated with stratosphere-troposphere separation ($2 \times 10^{14}$ molec cm$^{-2}$). We've clarified this in the revised manuscript.

**Response to comments by Referee # 2**

This manuscript analyzes $NO_x$ concentrations and the vertical distribution in the troposphere based on three aircraft campaigns - SEAC$^4$RS, DC3 & ATom – and four atmospheric chemistry models – GEOS-Chem, GMI, TM5 & CAMS. The authors present that measurements via LIF and P-CL overestimate $NO_2$ concentrations in the upper troposphere due to thermal interferences and can be better represented by PSS calculations from measured NO. $pNO_3^-$ photolysis as a missing $NO_x$ source in models is evaluated and is found to have a significant contribution, particularly over the oceans which improves the model performance for the ATom mission. Lightning and aircraft emissions are identified as main sources of NOx in the free troposphere, with individual contributions varying by latitude and season.

This manuscript is well written and presents an interesting and comprehensive analysis of free tropospheric $NO_x$ from measurements and models.

My main concerns are that the P-CL $NO_2$ measurements were not corrected for thermal interferences from e.g. MPN and $HNO_4$. This should be done and compared to the LIF measurements and the model results. I suggest some changes to the Figures, i.a. adding letters (a), (b), … to the subfigures for easier distinction, changing some of the axis scales and adding labels to all axis, but most importantly including error bars. Finally, I recommend the authors to elaborate in more detail how the $NO_2$ columns were calculated from the measurements and what the uncertainties are. You will find my detailed comments and questions below. Once these points are addressed properly, this paper would be a valuable contribution to the current literature.

We thank the reviewer for a thorough review of the manuscript and thoughtful comments. Our response to the main concerns and specific comments is as follows:

Main concerns:

*Correction to the P-CL $NO_2$ measurements:* The interferences in the P-CL $NO_2$ measurements have not yet been quantified and we do not have a method to correct them. This would need to be tackled through separate laboratory studies or instrument intercomparison studies. Besides, applying a correction for the thermal dissociation of species like $HNO_4$ and MPN based on their measured concentrations and their calculated dissociation fractions in the instrument may not enough. Such a correction was applied to the LIF $NO_2$ measurements, but this did not resolve the discrepancy of the measurements with the PSS-derived $NO_2$ concentrations. The residence times in the P-CL instrument (0.75 s) and the LIF instrument used in DC3 (0.5 s) are not that different, and we wouldn't expect the instrument temperature to be that different either. There is clearly a need to improve the P-CL $NO_2$ measurements, and we now stress this in the conclusions.

*Improvements to figures*: We have revised our figures to improve clarity by using more distinct colors, labeling subpanels wherever we refer to them individually, changing the scales of certain axes, and including error bars.

*Calculation of $NO_2$ columns*: We now describe how the $NO_2$ columns are calculated from the concentration profiles. See our response to comment #29 below.

**Major comments:**

1. **Lines 156 ff.: I assume this interference results from thermal decomposition of MPN. Could you go a bit more into detail? How was the correction determined? What's the temperature in the instrument and what is the residence time of the sample gas?**
   The correction method is fully described in Nault et al. (2015). We have cited that study and have added a brief description, including the instrument temperature and the residence time of the sample in the instrument.

2. **Lines 162 ff.: What's the residence time and the temperature in the photolysis cell (I assume the operation of the LEDs inevitably increases the temperature in the cell)? And how large is the resulting interference? Does thermal decay of $HNO_4$ play a role? From Bourgeois et al. (2022), I understand that the wavelength of the LEDs in the photolysis cell is 385nm – do you experience and correct for photolytic interference from HONO?**
   The residence time of the sample in the instrument was 0.75 s (Bourgeois et al. 2022). The temperature of the photolysis cell was not monitored, but there was little heating since the instrument uses low-power LEDs (Bourgeois et al. 2022). We have specified this in the revised manuscript.
   We do not how much $HNO_4$ dissociates in the instrument, but the results of this work suggest that all of it does (lines 336-339).
   There is photolytic interference from HONO (5% of the HONO mixing ratio), but it is small since HONO concentrations are less than 10 pptv in the free troposphere. We have specified this too in the revised manuscript.

3. **Table 1: It would be helpful to add the uncertainties of each measurement in the table (e.g. as a fifth column).**
   We now include the measurement uncertainties in the revised manuscript.

4. **Line 196: Do you perform a separate model run for the data along the flight track?**
   No, the model is sampled at the time and location of each flight while it is running. This is now clarified.

5. **Line 252: How is the factor of 2.39 determined?**
   It is based on the Na+ fraction in the standard seawater composition (Millero et al., 2008). We have clarified this now.

6. **Line 278: It looks like the DC3 NO observations show a minimum for the lowest altitude bin. Is this real and if yes, why? Or do you have a small number of observations at this altitude?**
   The DC3 NO measurements in the 0-1 km bin are not significantly different from measurements in 1-2 and 2-3 km bins considering the variability in each bin.

7. **Line 284: Do I understand correctly that the P-CL $NO_2$ measurements were not corrected for the MPN and $HNO_4$ interferences?**
   They have not been corrected, as discussed in our response to your main concerns above.

8. **Figure 1: I recommend adding letters (a)-(f) to the subfigures for easier distinction. Could you lower the upper x-limit for NO to e.g. 200 or 250 pptv – so the profile is better to see. I find it a bit confusing that black was chosen as the observation in the NO vertical profiles and for the PSS calculation in the $NO_2$ vertical profile. The LIF and P-CL observations are also hard to distinguish. I suggest using different colors and potentially decreasing the line width to prevent overlapping. Please add error bars to the modeled data, too.**
   We have changed the colors of the LIF and P-CL observations to improve clarity and have added error bars to the model data. We show the PSS $NO_2$ in black because it is a better

estimate of the $NO_2$ observations in the upper troposphere than the measurements. It also maintains consistency between Figures 1 and 4.

9. **Lines 307 ff.: So, the PSS is not entirely calculated from observations? - Maybe clarify in the legend of Figure 1.**
We have added the suggested clarification.

10. **Lines 312: It might be more accurate to use k(NO+CH$_3$O$_2$) as surrogate for k$_3$=k(NO+RO$_2$).**
Yes, and we have changed it in the revised manuscript, although this makes little difference in the free troposphere.

11. **Line 320: Does jNO2 increase with altitude and plays a role in this, too?**
There is relatively little change in $j_{NO_2}$ with altitude. We now state this in the revised manuscript.

12. **Lines 326 ff.: You can estimate the interference from HNO$_4$ and MPN with the temperature and the residence time in the instrument assuming first order decay (presented by Reed et al. (2016) and Nussbaumer et al. (2021)). If I see correctly you have measurements of both species available for SEAC$^4$RS & DC3 and can calculate the resulting artifact and correct the measurements. For ATom (if needed), you could estimate HNO$_4$ and MPN via PSS calculations. It could be helpful to compare your measurements with the PSS calculations for SEAC$^4$RS & DC3 to see whether using the PSS to calculate HNO$_4$ and MPN is a valid assumption. You can find the rate constants for the decay here: https://iupac-aeris.ipsl.fr/#.**
This was done for the LIF measurements, but as our results show this is not enough to improve the agreement of the $NO_2$ measurements with PSS. The interference would really need to be quantified through separate laboratory studies or instrument intercomparison studies.

13. **Line 329: I don't agree with the generalized assumption that HNO$_4$ and MPN dissociate with the same efficiency as NO$_2$. This largely depends on the temperature and the residence time. NO$_2$ is subject to photolytic dissociation by the LEDs in the converter (which is mostly temperature independent), while HNO$_4$ and MPN produce artifacts through thermal dissociation which is strongly temperature-dependent. MPN dissociates at lower temperatures compared to HNO$_4$, and therefore produces the NO$_2$ artifact at a higher efficiency than HNO$_4$.**
We agree. We meant to say that they dissociate completely in the instrument and are detected at the same efficiency as $NO_2$ (which is 40%). We have clarified this now.

14. **Line 332: How does it change the agreement between PSS NO/NO$_2$ and P-CL NO/NO$_2$ when accounting for the thermal interferences?**
We do not know how much interference there is in the P-CL $NO_2$ measurements from $HNO_4$ and MPN but our results suggest that these species dissociate completely in the instrument.

15. **Line 341: Do you observe fresh lightning peaks in your NO signal which support this statement?**
No, the storm had mostly dissipated by the time the sampling took place. We have revised this statement accordingly.

16. **Line 343: It would be helpful to also show the trace gas concentrations as a function of time of the measurement period. Does NO continuously decrease over the period? How long is the measurement period?**
Nault et al. (2016) have previously showed the $NO_x$ and $NO_y$ concentrations as a function of time and show a steady decrease in NO over the measurement period, which was about 2 hours. We have stated this in the revised manuscript and have cited this paper.

17. **Line 345: This likely looks different when correcting the P-CL NO$_2$ data as described above. Are the changes described for the NO$_2$ observations with increasing NO$_y$/NO molar ratio significant? It looks like the scatter of the 1-minute observations is**

**approximately the same as the increase/decrease of the NO$_2$ lines.**
The P-CL data were not corrected, as discussed above. The changes in NO$_2$ concentrations were not statistically significant and we have clarified this now.

18. **Line 348: Is [HO$_2$]=[RO$_2$] a valid assumption? What's the error arising from this assumption?**
We take [RO$_2$] = [HO$_2$] as an upper estimate. The PSS NO$_2$ concentrations are not too sensitive to this assumption about RO$_2$; they would be 10% lower if we assume [RO$_2$] to be half of the measured [HO$_2$].

19. **Equation (4): How did you calculate the ozone loss term via photolysis?**
The rate constant for the photolytic loss of ozone is calculated as:

$$k_{O_3 \to OH} = \frac{j_{O_3 \to O(^1D)} \, k_{O(^1D)+H_2O}}{k_{O(^1D)+M}} \frac{[H_2O]}{[M]},$$

$j_{O_3 \to O(^1D)}$ is the frequency of O$_3$ photolysis channel producing O($^1$D) and was measured on the flight. [H$_2$O] and [M] are calculated from meteorological observations on the flight. We have added this to the manuscript.

20. **Figure 2: Again, letters (a)-(c) for the subfigures would be helpful to follow in the text. I recommend using the same colors as in Figure 1 for the according measurements / calculations. Again, LIF NO$_2$ and P-CL NO$_2$ are hard to distinguish, and it is impossible to see the difference between the 1-minute observations. It could be helpful to use error bars instead. Please add error bars or the 1-minute data points for all species.**
To improve clarity we have removed the 1-minute observations and have added error bars instead.

21. **Lines 394 ff.: This seems a little bit like a circular argument to me. Wasn't the PPS NO$_2$ applied for the observations because the model identified a bias in the P-CL & LIF NO$_2$ measurements (Lines 281 ff.)? And now the model performance is evaluated in comparison to the PSS NO$_2$?**
The model performance is evaluated against the PSS NO$_2$ concentrations which are inferred from the NO (and other) measurements and are largely independent of the model (except for the use of modeled [RO$_2$], [BrO] and, for SEAC$^4$RS, [HO$_2$]).

22. **Line 403: How do these values compare to PSS calculations of MPN (production via CH$_3$O$_2$+NO$_2$ and loss via decay and photolysis)?**
Please refer to Figure 11 in Nault et al. (2015) for a comparison of MPN measurements to values from PSS, which shows reasonable agreement.

23. **Lines 411 ff.: This whole paragraph describes a method which is used in the subsequent section. I recommend shifting this paragraph to section 3.2.**
This is also relevant to the present section, which includes a comparison of the AMFs corresponding to the PSS and modeled NO$_2$ profiles for SEAC$^4$RS and DC3.

24. **Figure 3 and 4: Consider using letters for the subfigures. Why was the log scale chosen for the x-axis? I would find it more straight forward with a linear scale and it would also be easier to compare to Figure 1. Please add horizontal error bars for all traces.**
The log scale was used to show the values in the lower troposphere clearly, since NO concentrations there are much lower than in the upper troposphere. We have clarified this in the figure caption now. We have also added error bars to the GEOS-Chem profiles in the revised manuscript, but not to all profiles to avoid unnecessary clutter in the figure.

25. **Lines 447 ff.: How does this compare to the satellite measurements?**
    This is generally consistent with satellite measurements. We have stated this now.
26. **Line 471: Could you show a vertical profile of $pNO_3^-$ somewhere - either in the manuscript or the supplement?**
    We have added this figure as a supplement.
27. **Figure 5: Please add labels to the axis and the color bar.**
    We have made the requested changes to the figure.
28. **Line 533: I cannot follow this argument. It looks like measured $pNO_3^-$ is much larger than the modeled values. Please clarify.**
    We have clarified this now and have added a figure showing a comparison of the simulated and observed $pNO_3^-$ concentrations.
29. **Lines 552 ff.: How are the PSS $NO_2$ column density and the corresponding AMF determined? Could you provide a more detailed procedure? From Eq. (5) it looks like the viewing geometry of the satellite is required to determine the AMF. How does this apply to PSS or (in situ) measured values?**
    We have described this in the revised manuscript. The viewing geometry is accounted for by $AMF_G$.
30. **Lines 562 ff.: Considering a column density of $\sim 1.3 \times 10^{14}$ molec/cm² for GMI, isn't a difference of $\sim 1 \times 10^{14}$ molec/cm² quite large ($\sim 80\%$)?**
    We hadn't written this clearly. We were comparing this value to the uncertainty in the stratosphere-troposphere separation step, which is $\sim 2 \times 10^{14}$ molec/cm². We have clarified this now.
31. **Figure 6: What are the errors on these values?**
    We have added error bars to the values in the revised manuscript.
32. **Line 582: Do these values refer to the annual mean over all layers? Looking at the $NO_x$ changes in Figure 7 (up to 400%), the 9% value probably has a large uncertainty. It could be helpful to add the uncertainty e.g. as the $1\sigma$ standard deviation.**
    The values refer to the change in the tropospheric mass of NOx, OH, and ozone (sum of the mass of the species in each model grid cell in the troposphere). The relative increase in the mass of these species in some areas is indeed much higher than the relative increase in the tropospheric mass. We point this out in the revised manuscript.
33. **Figure 7: Why do you show two plots with % changes and one with ppb changes?**
    We show the ozone changes in units of ppbv because the values are more meaningful and are readily comparable to results from other studies, such as Wang et al. (2021), which we discuss in the manuscript.
34. **Lines 615 ff.: Please state the $1\sigma$ values (or something comparable) for all averages.**
    We now state the $1\sigma$ values where they are most relevant.
35. **Line 672: It looks like for the 30-60°N oceans and the U.S., absolute aircraft emissions are slightly larger in February compared to August, e.g. at 12km altitude for the U.S. $\sim 70$pptv $NO_x$ in February and $\sim 30$-$40$pptv $NO_x$ in August. Is this significant? Do you have an explanation for that?**
    There is little seasonal variation in the aircraft $NO_x$ emissions in GEOS-Chem. The lower $NO_x$ concentrations in summer most likely reflects the shorter $NO_x$ lifetime against oxidation by OH.
36. **Figure 9: Please add error bars, e.g. $1\sigma$ from averaging at each altitude bin. Please consider using letters (a) and (b) instead of referring to the right and the left panel.**
    We have made the suggested changes to the figure.
37. **Lines 739 ff.: I am not sure I can completely follow this argument. So, the right panel from Figure 9 show the summed contribution at each altitude plus everything that's**

**above up to 12km? Why do the winter profiles not show 100% at ground level?**
Yes, it shows summed contribution at each altitude plus everything above up to 12 km. Thank you for noticing that the winter profile didn't reach 100% at the ground level. There was an error in the plot (the bottommost value was left out), which we have fixed now and the profiles reach 100% at ground level.

**Minor comments:**

**Line 346: shows**
Corrected, as suggested.

**Line 613: consider replacing 'worsening' with 'reducing'**
Replaced, as suggested.

We now include the following recommendation in the Conclusions:

> There is a need to improve $NO_2$ measurements in the free troposphere. At present, $NO_2$ concentrations inferred by applying PSS to NO and other measurements provide a better estimate of free tropospheric $NO_2$ than the direct measurements, and we use them as basis for evaluating the models.

As stated in our response to comment #19 below, we can't be more specific about our recommendations based on the results of this work alone. These issues need to be tackled in future studies.

**Finally, the authors should discuss recommendations for observing emissions trends given the importance of the free tropospheric contribution to columns, for example by only looking at regions with column amounts over a certain threshold. They haven't quantified for us how important the free tropospheric background is to the column over cities and whether an approach removing columns with a significant background contribution could help to clarify trends in surface NOx emissions.**
The last paragraph of Section 3.5 specifically addresses this. For example, we point out that "the free tropospheric contribution would be small over major cities, where summertime $NO_2$ column densities exceed $3 \times 10^{15}$ molec cm$^{-2}$." We also give examples of studies focused on urban emissions that do explicitly account for the free tropospheric background. In light of the reviewer's comment, we have now added the point about the free tropospheric contribution being not as much of a problem for urban areas to the Conclusions.

**Minor comments**

1. Line 129 – This statement is confusing "Silvern et al. (2018) showed that using the observed NO2 vertical profile from SEAC4RS in the NASA NO2 column retrieval for the OMI satellite instrument decreases the retrieved NO2 columns over the southeastern US by 30%, suggesting the possibility of a systematic bias in the NO2 column retrievals." You have just told us that NO2 measurements have interferences, so likely the observed profile is biased, not the retrieval. Please clarify.
   This statement was contradictory, and we have removed it from the manuscript.
2. In Bradshaw et al., 1999, observations matched photostationary steady state, using a "a highly modified photofragmentation two-photon laser-induced fluorescence (PF-TP-LIF) instrument." Could you explain how these modifications differ from current LIF techniques and provide some guidance on whether the modifications from Bradshaw et al., 1999 should be employed in the future to better measure NO2? Similarly, can you explain why the MPN corrections in the SEAC4RS LIF observations were insufficient?
   The main advantage of the LIF instrument is that it measures $NO_2$ directly, whereas the Bradshaw et al. (1999) instrument photolyzes $NO_2$ to NO and measures the NO signal and is thus more like the P-CL instrument. The modifications that the reviewer is likely referring to are the unusually large inlet size and the high flow rates used by Bradshaw et al. (1999) in order to reduce the dissociation of $HNO_4$ and MPN (and PAN) through collisions with the inlet walls. It is

possible that heterogeneous dissociation of HNO$_4$ and MPN contributes to the interference in the LIF measurements, which the correction for thermal decomposition would not account for. We now state this in the revised manuscript.

As stated in our manuscript, the problem with the MPN corrections is that there is high uncertainty in the MPN thermal dissociation rate constant and the MPN measurements and there could also be wall-mediated dissociation.

3. **Line 248 – How do you justify that sea salt is internally mixed with sulfate/ammonium/nitrate in the fine mode?**
   The model treats fine mode aerosols as well-aged particles that have undergone internal mixing through coagulation and cloud processing. The model also includes the formation and uptake of sulfate and nitrate on sea salt aerosols, titrating aerosol alkalinity and displacing chloride. We have clarified this in the revised manuscript.

4. **Line 390 – Please clarify. Is this 'unheated' channel the one you are saying still has a measurement bias, presumably because the channel is still warmer than ambient temperatures due to being on the inside of the aircraft?**
   The unheated NO$_2$ channel is indeed at the aircraft cabin temperature and prone to interference from thermal dissociation MPN and HNO$_4$ when sampling in the much colder free troposphere, for which the NO$_2$ measurements were corrected (Nault et al., 2015) as described in Section 2.1. We have clarified this in the revised manuscript.

5. **Line 410 – I am curious if your simulations with photolysis of particulate nitrate produced appreciable HONO during SEAC4RS/DC3 as SENEX observations showed HONO outside of direct sources was negligible (< 15 ppt) https://agupubs.onlinelibrary.wiley.com/doi/abs/10.1002/2016JD025197.**
   The HONO concentrations in the BL in our simulation with pNO$_3^-$ photolysis for SEAC$^4$RS and DC3 was about 4 pptv.

6. **Line 430 – How does this finding change the conclusion of Travis et al., 2016 that the model NO2 columns were inconsistent with other constraints on emissions?**
   Our results are consistent with those of Travis et al. (2016), who suggested an overestimate of NOx emissions in the US EPA NEI. Our manuscript states the following:
   > NOx emissions are likely overestimated in the US EPA NEI inventory used in our simulations (Travis et al., 2016), which explains the NO$_2$ overestimate in the boundary layer…"

7. **Line 435 – Does it really make sense to discuss a 'global background of NO2' given its short lifetime of only hours? Also, "background" is sometimes used to mean non-anthropogenic. Possibly consider different terminology.**
   Good point! We now use NO$_2$ in the free troposphere and remote regions instead of a 'global background NO$_2$', and wherever we use "background" NO$_2$ we refer to it as the "free tropospheric background."

8. **Fisher et al., 2018 (https://agupubs.onlinelibrary.wiley.com/doi/10.1029/2018JD029046) suggest that the underestimated NO could be direct emissions from seawater. Discuss whether this is a possible alternative to photolysis of particulate nitrate.**
   We have included a discussion of this point in the revised manuscript.

9. **Line 474 – As you are also simulating HNO3 and PAN, it would be more consistent to compare your own simulations to HNO3 and PAN, and then you could avoid a citation to a presentation (Zhai et al., 2022) rather than peer-reviewed literature. The authors should also show comparisons with observed particulate nitrate and show whether photolysis improves or degrades that simulation.**
   We now include a figure in the supplement showing comparisons of the observed and simulated HNO$_3$, PAN, and pNO$_3^-$.

10. **Line 474 – How do you know there is no overestimate of NOx sinks? You haven't shown us this. Or do you just mean NO2 + OH is not overestimated which is why you cite Travis et al., 2022 for OH?**
Fair point. We do have good support for the $NO_2+OH$ sink based on the comparison of GEOS-Chem OH concentrations with the ATom measurements by Travis et al. (2020). The heterogeneous $NO_x$ chemistry in the model reflects the state of the science, but while there is uncertainty in that chemistry, those processes are important mostly in the midlatitudes in winter/spring. We have clarified this in the revised manuscript.

11. **Line 476 – Also Henderson et al., 2012 - https://acp.copernicus.org/articles/12/653/2012/ and Seltzer et al., 2015 https://acp.copernicus.org/articles/15/5973/2015/acp-15-5973-2015.pdf.**
We now cite these papers.

12. **Line 585 – Can you discuss again the biases in your GEOS-Chem simulation of PAN in the context of this discussion? Is PAN underestimated in the MBL?**
We now include a figure in the supplement comparing the GEOS-Chem and observed PAN concentrations in ATom, which shows no underestimate in the modeled PAN in the MBL.

13. **Line 620 – This conclusion is without addressing model underestimates of OH reactivity. Please discuss how you would expect model ozone to change if OH reactivity was increased in the model.**
The ozone bias in the model could be even larger if the OH reactivity was higher, but it doesn't change our conclusion that including $pNO_3^-$ photolysis will improve the ozone simulation. If anything, we would need to include such a process in the model even more.

14. **Line 694 – Why would estimating free tropospheric concentrations in February be beneficial? What would be the application for such an estimate?**
It is important to examine free tropospheric $NO_2$ in the winter in light of the results of Marais et al. (2018), who found that GEOS-Chem underestimates free tropospheric $NO_2$ in winter compared to estimates derived from OMI observations using the cloud-slicing method, implying potential errors in NOx sources/chemistry/transport. Instead, our results suggest that this likely reflects uncertainties in the cloud-sliced products. This is discussed in the subsequent lines.

15. **Line 718 – What are the differences in the reaction rates for NO + O3 and NO2 + OH + M in GMI vs. GEOS-Chem?**
Ozone concentrations are very similar in the two models, but OH concentrations are somewhat lower in GMI, which would lead to a slightly longer NOx lifetime in GMI. We now state this in the manuscript.

16. **Figure 1 – Is there a reason to put NO on the same scale as NO2? It would be easier to see the discussed NO biases if the scale was reduced.**
We have changed the scale of the NO plot for better visibility of the data.

17. **Line 732 – Is there a reason this decrease would impact only the free troposphere and not the boundary layer?**
Decreasing both the free troposphere and the boundary layer by half would have no effect on the AMF since the shape factor would remain the same. We clarify this in the revised manuscript.

18. **Line 765 – I assume then that the higher spatial resolution of TEMPO will allow for a better examination of trends in emissions moving forward as it will be less sensitive to FT background. Also, why haven't studies only focused over urban regions to look at trends?**
TEMPO will indeed better resolve urban NOx concentrations than OMI, but several studies

have used OMI (and TROPOMI) observations to derive emissions in urban areas and have adequately accounted for the free tropospheric column (examples cited in last paragraph of Sect. 3.5).

19. **Line 776 – Can you clarify exactly what needs to be done for future NO2 measurements? Reconsider the amount of MPN that might interfere? Make the modifications described in Bradshaw et al., 1999?**

    It is not possible to be specific about what needs to be done based on results of this work, because it is not clear whether it is just that the MPN correction is underestimated, or whether there are other (unknown) $NO_2$ reservoir species that could interfere (Silvern et al, 2018), or there are wall-mediated losses that are not accounted for (Bradshaw et al. 1999). This needs to be tackled in future work. Our recommendations about how the $NO_2$ measurements should be treated are now included in the conclusions.

20. **Line 784 – This again makes me curious about peroxy nitrates (PAN etc) as well, and whether the model underestimates NOx transport that could be part of the underestimate in ATom NO. You say on line 299 that PAN is small, but also the perturbation in NO needed is small.**

    We have included a figure comparing the simulated and observed PAN in ATom, and we find general agreement for PAN in the tropics and southern midlatitudes. There is ~30% underestimate in the northern midlatitudes, but this is not enough to account for the factor of 2 underestimate in NO. We have included this discussion in the revised manuscript.

21. **Line 805 – But again, how would you expect the ozone response to change if OH reactivity was increased?**

    Please see our response to comment #13. It doesn't change our conclusion about the effect of $pNO_3^-$ photolysis on ozone in the model.

22. **Line 813 – Is there any reason to suggest focusing analysis of NOx trends/emissions on regions where the NOx column is high enough to have low uncertainty from the large free tropospheric background?**

    The free troposphere is indeed less of a problem in urban areas. We have included this in our conclusions.

*References:*

[revised manuscript text omitted]

---

## Author Response (AR2)

**Response to referee comments**

The authors addressed most of my comments in their revised manuscript. I only have some few comments left. Once these are addressed, I recommend this paper for publication.

We thank the reviewer for their constructive comments.

It would be very helpful if (for next time) the authors could state the lines of changes in the new manuscript in the 'response to the reviewers' document and cite the corrections made.

Thermal interferences:
Point 2: Even low power LEDs could potentially increase the temperature in the converter significantly. Maybe you could measure it retrospectively in the lab. If this is not possible, I recommend doing a small sensitivity study assuming for example temperatures of a) 20°C b) 40°C and c) 60°C in the converter and calculating the resulting decay of HNO4 and MPN and the resulting NO2 interference. The authors have all the necessary measurements for this calculation and it would make the CLD NO2 data more accurate and more valid to compare to other NO2 data.

We looked into this further following the reviewer's comments. The temperature of the LEDs used for photolysis was measured and based on that we estimate that temperature in the photolysis cell (converter) was 20–30°C, which would result in a 100% dissociation of MPN and 30-40% dissociation of $HNO_4$. We do not correct for this interference because MPN measurements were not available for ATom, and while available for SEAC[4]RS and DC3 have a high uncertainty (40%, Nault et al. 2015). We now include the following in the revised manuscript at Line 162:

"Interference in the $NO_2$ measurement from $HNO_4$ and MPN is estimated to be 30–40% for $HNO_4$ and 100% for MPN based on an estimated photolysis cell temperature of 20–30ºC and the residence time of air in the cell of 0.75 s during ATom (Bourgeois et al., 2022). The P-CL $NO_2$ measurements are not corrected for this interference."

According to Bourgeois et al., the volume of the converter is ~51 cm³ with a flow of ~1.03 SLM and a pressure of ~280 hPa. This results in a residence time of > 0.8 s alone in the converter. Therefore, a residence time of 0.75 s in the whole instrument seems to be incorrect. Please double check the stated value.

We double checked this value. The residence time in the photolysis cell (converter) is 0.75 s (Bourgeois et al. 2022). This represents the bulk of the residence time in the instrument (the residence time in the rest of the instrument is 0.1-0.2 s), and the photolysis cell is where most of $HNO_4$ and MPN decay would take place. The reviewer may not have considered the temperature in their calculation of the residence time.

The authors assume that all of MPN and HNO4 decay in the instrument. However, under the conditions in the converter, the temperature would need to be >80°C for a full decay of HNO4, according to my calculations. This seems unlikely. Please provide some calculations for your assumption.

We didn't assume this a priori. It was based on the consistency between the ratio of $NO/NO_2^*$ ($NO_2^* \equiv NO_2 + HNO_4 + MPN$) in GEOS-Chem the $NO/NO_2$ ratio for the P-CL $NO_2$ measurements. However, to avoid confusion we have removed this statement from the revised manuscript. The modified sentence is as follows (Line 334):

"The P-CL $NO_2$ instrument has significant interference from the dissociation of $HNO_4$ and MPN (Reed et al., 2016; Nussbaumer et al., 2021; Bourgeois et al., 2022), and we find that the ratio of $NO/NO_2^*$ ($NO_2^* \equiv NO_2 + HNO_4 + MPN$) in GEOS-Chem matches the $NO/NO_2$ ratio for the P-CL $NO_2$ measurements."

Figure 1: There are some error bars missing. I would still find it much easier to follow the text if the authors decided to label the subpanels.

We have added labels to the subpanels and refer to them in the text.

Point 17: I cannot find this statement in the revised manuscript.

The statement was added on Line 356: "But there was relatively little change in the $NO_2$ concentrations."

Point 19: For your stated equation, [M] should represent [O2], [N2] and [H2O]. But the individual rate constants of O(1D) with these species are different. How did you determine k(O(1D)+M)?

$k_{O(^1D)+M}$ is the weighted-average reaction rate constant of $O(^1D)$ with $N_2$ and $O_2$ (Seinfeld and Pandis, 2016). We have added this now to the revised manuscript (Line 380).

Figure 2: There are still some error bars missing.
We have added the missing error bars in the revised figure.

Figure 4: I can see that error bars for all species make the graph a bit chaotic, but you could show it in the supplement for the individual species instead or add a description about the error range.
We have added a figure in the supplement showing the error bars for the remaining species.

Reference:
Bourgeois, I., Peischl, J., Neuman, J. A., Brown, S. S., Allen, H. M., Campuzano-Jost, P., Coggon, M. M., DiGangi, J. P., Diskin, G. S., Gilman, J. B., Gkatzelis, G. I., Guo, H., Halliday, H. A., Hanisco, T. F., Holmes, C. D., Huey, L. G., Jimenez, J. L., Lamplugh, A. D., Lee, Y. R., Lindaas, J., Moore, R. H., Nault, B. A., Nowak, J. B., Pagonis, D., Rickly, P. S., Robinson, M. A., Rollins, A. W., Selimovic, V., St. Clair, J. M., Tanner, D., Vasquez, K. T., Veres, P. R., Warneke, C., Wennberg, P. O., Washenfelder, R. A., Wiggins, E. B., Womack, C. C., Xu, L., Zarzana, K. J., and Ryerson, T. B.: Comparison of airborne measurements of NO, $NO_2$, HONO, $NO_y$, and CO during FIREX-AQ, Atmos. Meas. Tech., 15, 4901–4930, https://doi.org/10.5194/amt-15-4901-2022, 2022.

Nault, B. A., Garland, C., Pusede, S. E., Wooldridge, P. J., Ullmann, K., Hall, S. R., and Cohen, R. C.: Measurements of $CH_3O_2NO_2$ in the upper troposphere, Atmos Meas. Tech., 8, 987–997, https://doi.org/10.5194/amt-8-987-2015, 2015.

Seinfeld, J. H. and Pandis, S. N.: Atmospheric chemistry and physics: from air pollution to climate change, Third edition., John Wiley & Sons, Hoboken, New Jersey, p.177, 2016.